# Nonequilibrium steady states in the Floquet-Lindblad systems: van Vleck's high-frequency expansion approach

Tatsuhiko N. Ikeda[1*], Koki Chinzei[1], Masahiro Sato[2]

**1** Institute for Solid State Physics, University of Tokyo, Kashiwa, Chiba 277-8581, Japan
**2** Department of Physics, Ibaraki University, Mito, Ibaraki 310-8512, Japan
\* tikeda@issp.u-tokyo.ac.jp

November 10, 2021

## Abstract

Nonequilibrium steady states (NESSs) in periodically driven dissipative quantum systems are vital in Floquet engineering. We develop a general theory for high-frequency drives with Lindblad-type dissipation to characterize and analyze NESSs based on the high-frequency (HF) expansion with linear algebraic numerics and without numerically solving the time evolution. This theory shows that NESSs can deviate from the Floquet-Gibbs state depending on the dissipation type. We show the validity and usefulness of the HF-expansion approach in concrete models for a diamond nitrogen-vacancy (NV) center, a kicked open XY spin chain with topological phase transition under boundary dissipation, and the Heisenberg spin chain in a circularly-polarized magnetic field under bulk dissipation. In particular, for the isotropic Heisenberg chain, we propose the dissipation-assisted terahertz (THz) inverse Faraday effect in quantum magnets. Our theoretical framework applies to various time-periodic Lindblad equations that are currently under active research.

# 1   Introduction

Periodically driven quantum systems have seen a resurgence of interest motivated by laser technology advancement and theoretical developments [1–5]. Such systems are theoretically described by the Floquet theorem [6,7], which enables us to study nonequilibrium states of matter systematically [8,9]. An important application is the Floquet engineering, i.e., artificially creating useful functionalities of physical systems by designing an appropriate driving protocol. Floquet engineerings for various systems have been proposed theoretically and realized experimentally: the dynamical localization [10–14], Floquet topological states [15–19], Floquet time crystals [20–22], the inverse Faraday effect [23–25], etc.

Despite those extensive studies, most of their theoretical analyses neglect the dissipation effect and focus mainly on well-controlled artificial systems and ultraclean materials. Considering dissipation is important in two ways. First, most physical systems such as generic materials contact their environment, and dissipation is not negligible. Second, without dissipation, the Floquet-engineered states are eventually broken by injected heat, and the system becomes the featureless infinite-temperature state [26–28]. Thus, in isolated systems, the Floquet engineering is usually considered in the Floquet prethermalization regime of finite time window [29–33]. In dissipative systems, nonequilibrium steady states (NESSs), where the energy injection by the periodic driving is balanced with the energy dissipation, are established

and give us the opportunity to the long-lived Floquet engineering.

There have been several approaches to analyze NESSs in periodically driven dissipative systems. The nonequilibrium Green function method is a successful approach and particularly useful in electron systems coupled to several leads [34–37]. Several systems have been analyzed with this approach, such as the Floquet topological insulators [38], strongly-correlated electrons [39], and so on. Quantum master equation [40, 41] is another useful formulation to analyze various periodically-driven dissipative systems and has been applied to periodic thermodynamics [42–44], Bose-Einstein condensation [45], Floquet-band occupation [46], and so on. Fermi's golden rule is a similar technique to define the NESS [47].

Among quantum master equations, the Floquet-Lindblad equation (FLE) is a powerful and flexible description for periodically-driven systems subject to Markovian dissipation [48–51]. The FLE is an extension of the Lindblad (or Gorini–Kossakowski–Sudarshan–Lindblad [52, 53]) for time-periodic systems and has nice properties such as the complete positivity and trance-preserving condition. However, it is generally hard to solve each FLE and find the NESS except for some special cases [54–56]. Recently, based on the high-frequency expansion method, a part of the present authors [57] have analytically solved NESS in a phenomenological FLE. However, there are two major remaining issues to be solved. First, that work assumed that the dissipation is time-independent and satisfies the detailed-balance condition, but dissipation without these properties has also attracted attention in the literature. Second, the NESS was solved at the leading-order approximation for $1/\omega$ ($\omega$ is the driving frequency), and the systematic extension to higher orders has not been studied.

In this paper, we extensively study the NESS in the FLE for high-frequency drives. For this purpose, we formulate how to characterize and obtain the NESS using the high-frequency (HF) expansion for the Liouvillian (Lindbladian) from the van Vleck point of view. Compared to the Floquet–Magnus viewpoint [50], the van Vleck approach involves fewer terms and enables us to study NESSs, including the micromotion. Following the general formulation, we apply this method to example models: an effective three-level model for a nitrogen-vacancy (NV) center in diamond, a periodically driven topological XY spin chain, and the Heisenberg spin chain under a circularly polarized magnetic field. The HF-expansion approach enables us to analyze the NESS systematically by numerically obtaining an eigenstate of the effective Liouvillian without resorting to the direct numerical integration of the FLE. From our formalism, we also reveal the condition that the NESS approaches or deviates from the Floquet-Gibbs state.

The rest of this paper is organized as follows. In Sec. 2, we introduce the FLE and formulate our problem of finding the NESSs that we address in this paper. In Sec. 3, we develop the van Vleck HF expansion for Liouvillians. Importantly, we generally prove that a zero-mode exists for the effective Liouvillian at each order of the HF expansion. The zero mode is shown to correspond to the NESS in Sec. 4, where general aspects of the NESS are discussed within the FLE. In Sec. 5, we apply the HF expansion methods to time-independent dissipators, including two models. We first discuss an open XY spin chain under a periodic drive and boundary dissipation and analyze its topological phase diagram together with its stability against dissipation. We second analyze an effective model for the NV center in diamonds, demonstrating how the HF-expansion method gives the NESS accurately. In Sec. 6, we apply the HF expansion to a class of time-dependent dissipators derived from the system-bath coupling and the rotating-wave approximation (RWA). We derive such FLEs with a slight generalization of previous studies in that one allows the Floquet quasienergies to be degenerate and discuss the conditions for the NESS being approximated by the Floquet-Gibbs state. We then apply these general discussions to two concrete example models: the

effective model for the NV center and the Heisenberg spin chain under a circularly polarized magnetic field. For the NV center model, unlike with the time-independent dissipator, we show that the relationship between the driving frequency and the bath spectral cutoff brings about a nontrivial effect on the NESS. Namely, when the photon energy $\hbar\omega$ is below the cutoff, some photon-exchange processes are active, and the NESS cannot be described by the Floquet-Gibbs state. Even in this case, the HF expansion is valid and enables us to obtain the NESS with the systematic improvement of accuracy. For the Heisenberg chain, we first provide a microscopic theory for the dissipation-assisted inverse Faraday effect in magnetic insulators, i.e., the emergence of time-averaged magnetization due to the circularly polarized ac magnetic field. Unlike related studies, the magnetization is activated not by magnetic anisotropy of the Hamiltonian (the Heisenberg spin chain is isotropic) but by dissipation, i.e., the system-bath coupling. Finally, in Sec. 7, we summarize our work and present some outlooks.

## 2 Floquet–Lindblad Equation (FLE)

In this section, we introduce the most general form of the Floquet–Lindblad equation (FLE) that we study in this work and formulate our problem.

Let us consider the quantum master equation

$$\frac{\mathrm{d}}{\mathrm{d}t}\rho(t) = \mathcal{L}_t[\rho(t)]. \tag{1}$$

Here, $\rho(t)$ is the density operator describing the quantum state of the system of interest, and $\mathcal{L}_t$ is the Liouvillian superoperator generating dynamics. We use the term "Liouvillian" for the right-hand side of Eq. (1) by analogy with the Liouville equation in classical mechanics. Throughout this paper, we use calligraphic symbols for superoperators. We suppose that the Liouvillian is periodic in time: $\mathcal{L}_{t+T} = \mathcal{L}_t$, where $T$ is the period. In the following, we shall use the corresponding angular frequency $\omega \equiv 2\pi/T$.

We suppose that the Liouvillian is of Lindblad (or GKSL [52, 53]) form. Namely, the Liouvillian consists of the Hamiltonian and dissipator parts:

$$\mathcal{L}_t(\rho) = \mathcal{H}_t(\rho) + \mathcal{D}_t(\rho), \tag{2}$$

with

$$\mathcal{H}_t(\rho) \equiv -\mathrm{i}[H(t), \rho]. \tag{3}$$

Here, $H(t) = H(t + T)$ is the time-dependent Hamiltonian (involving the Lamb shift contribution [40]) in general). The dissipator is described by the jump operators $L_\alpha(t)$ as

$$\mathcal{D}_t(\rho) = \sum_\alpha \left[ L_\alpha(t)\rho L_\alpha^\dagger(t) - \frac{1}{2}\left\{ L_\alpha^\dagger(t)L_\alpha(t), \rho \right\} \right]. \tag{4}$$

We assume that $\mathcal{D}_t$ is also periodic: $\mathcal{D}_{t+T} = \mathcal{D}_t$. Under this assumption, we further assume that each $L_k(t)$ is also periodic[1]. In the following, we call $\mathcal{L}_t$ as the Lindbladian when we

---

[1]This assumption can be slightly relaxed as follows: For each $\alpha$, there exists $\theta_\alpha(t) \in \mathbb{R}$ such that $L_\alpha(t) = e^{\mathrm{i}\theta_\alpha(t)}L_\alpha^P(t)$ with periodic part $L_\alpha^P(t + T) = L_\alpha^P(t)$. In such cases, we can replace $L_\alpha(t)$ by $L_\alpha^P(t)$ without changing $\mathcal{D}_t$ since the nonperiodic phase factors $e^{\mathrm{i}\theta_\alpha(t)}$ cancel between $L_\alpha(t)$ and $L_\alpha^\dagger(t)$. Thus, under this assumption, we can assume $L_\alpha(t)$ are periodic without loss of generality.

emphasize that it is a Liouvillian of Lindblad form. We call the quantum master equation (1) generated by a time-periodic Lindbladian the FLE.

The periodicity in time of the FLE enables systematic analysis by Fourier expansions. We Fourier-expand the Hamiltonian and dissipator superoperators as

$$\mathcal{H}_t = \sum_m \mathcal{H}_m e^{-im\omega t}; \quad \mathcal{D}_t = \sum_m \mathcal{D}_m e^{-im\omega t}. \tag{5}$$

Here, the Fourier components $\mathcal{H}_k$ are simply given by

$$\mathcal{H}_m(\rho) = -i[H_m, \rho], \tag{6}$$

where $H_m$ are defined in $H(t) = \sum_m H_m e^{-im\omega t}$. In contrast, $\mathcal{D}_m$ are a little more complicated because it is not linear in $L_\alpha(t)$ and $L_\alpha^\dagger(t)$. Fourier expanding them as

$$L_\alpha(t) = \sum_m L_{\alpha,m} e^{-im\omega t}; \quad L_\alpha^\dagger(t) = \sum_m L_{\alpha,m}^\dagger e^{+im\omega t}, \tag{7}$$

we have

$$\mathcal{D}_m(\rho) = \sum_{\alpha,n} \left[ L_{\alpha,m-n}\rho L_{\alpha,n}^\dagger - \frac{1}{2}\left\{ L_{\alpha,n}^\dagger L_{\alpha,m-n}, \rho \right\} \right]. \tag{8}$$

We note that $L_\alpha(t)$ is not necessarily Hermitian, and $L_{\alpha,-m} \neq L_{\alpha,m}^\dagger$ in general.

The formal solution of the FLE (1) is represented by the propagator $\mathcal{V}(t,t')$ as

$$\rho(t) = \mathcal{V}(t,t')\rho(t') \tag{9}$$

with

$$\mathcal{V}(t,t') = \exp_+\left( \int_{t'}^t \mathcal{L}_s ds \right), \tag{10}$$

where $\exp_+$ denotes the time-ordered exponential. One merit of assuming that $\mathcal{L}_t$ is a Lindbladian at each time $t$ is that the propagator $\mathcal{V}(t,t')$ is guaranteed to be a completely positive and trace preserving (CPTP) map [52,53]. Thus, $\rho(t)$ is qualified as a density operator at any $t$ during time evolution, which is not guaranteed in other master equations such as the Redfield equation [58,59].

Finally, we discuss how the NESS is characterized in the FLE, following Ref. [60]. To this end, we take an initial time $t = 0$ and consider a long-time evolution to $t$. To utilize the periodicity $\mathcal{L}_{t+T} = \mathcal{L}_t$ and hence $\mathcal{V}(t+T, t'+T) = \mathcal{V}(t,t')$, it is useful to denote $t = t_0 + \ell T$, where $0 \leq t_0 < T$ and $\ell \in \mathbb{Z}$. Together with the property $\mathcal{V}(t,t') = \mathcal{V}(t,t'')\mathcal{V}(t'',t')$ for any $t''$, we have

$$\rho(t) = \mathcal{V}(t_0, 0)\mathcal{V}_F^\ell \rho(0), \tag{11}$$

where $\mathcal{V}_F \equiv \mathcal{V}(T, 0)$ is the one-cycle propagator. Being a CPTP map, $\mathcal{V}_F$ is known to have an eigenvalue 1, and we let the corresponding eigenstate be $\eta$. Supposing, for simplicity, that all the other eigenvalues have absolute values less than 1 (see Ref. [61] for a sufficient condition for this), we obtain the long-time behavior $\rho(t) \to \mathcal{V}(t_0, 0)\eta$ as $t \to \infty$. This is the NESS solution periodic in time, and $\mathcal{V}(t_0, 0)$ is called the micromotion within a period. Although the NESS is thus obtained from the one-cycle and micromotion parts of the propagator, it is difficult to obtain them analytically and a hard task to do them numerically.

# 3   van Vleck high-frequency expansion of Liouvillian

The high-frequency (HF) expansions offer systematic ways for looking into the propagator analytically when $\omega$ is large enough. As developed for isolated systems, there are, at least, three versions: the Floquet–Magnus [62], van Vleck [8], and Brillouin–Wigner [9]. These three look different but are related to each other by appropriate transformations [8, 9]. The Floquet–Magnus approach has been generalized to the FLE in the literature [50, 63]. In this section, we generalize the van Vleck approach to the FLE, which will be useful for analyzing the NESS in the following.

Before discussing the FLE, we briefly review the van Vleck HF expansion in isolated systems. In isolated systems, one tries to solve the Schrödinger equation $\frac{\mathrm{d}}{\mathrm{d}t}\left|\psi(t)\right\rangle = -\mathrm{i}H(t)\left|\psi(t)\right\rangle$ for a periodic Hamiltonian $H(t)$. The formal solution is given by $\left|\psi(t)\right\rangle = U(t,t')\left|\psi(t')\right\rangle$ with $U(t,t')$ being the unitary propagator from $t'$ to $t$. The HF expansion is derived from the following decomposed form of $U(t,t')$ [8, 9].

$$U(t,t') = \mathrm{e}^{-\mathrm{i}K(t)}\mathrm{e}^{-\mathrm{i}H_{\mathrm{eff}}(t-t')}\mathrm{e}^{\mathrm{i}K(t')}, \tag{12}$$

where $H_{\mathrm{eff}}$ is a time-independent effective Hamiltonian, $K(t) = K(t+T)$ is a periodic Hermitian operator, and $\mathrm{e}^{\mathrm{i}K(t)}$ is called the micromotion operator. With the series expansion, $K(t) = \sum_{k=1}^{\infty} K^{(k)}(t)$ and $H_{\mathrm{eff}} = \sum_{k=0}^{\infty} H_{\mathrm{eff}}^{(k)}$, we obtain each of $K^{(k)}$ and $H_{\mathrm{eff}}^{(k)}$ order by order, where $K^{(k)}$ and $H_{\mathrm{eff}}^{(k)}$ are $O(\omega^{-k})$. First few terms are given in Appendix A.

Now we generalize the van Vleck HF expansion to the FLE in open systems. To this end, we invoke the formal analogy between the FLE (1) and the Schrödinger equation. Although there is a difference between the operator $-\mathrm{i}H(t)$ and the superoperator $\mathcal{L}_t$, these are both linear operators. Thus, the derivation of the HF expansion goes in parallel by the formal substitution $H(t) \to \mathrm{i}\mathcal{L}_t$. More concretely, we put the following ansatz:

$$\mathcal{V}(t,t') = \mathrm{e}^{\mathcal{G}_t}\mathrm{e}^{\mathcal{L}_{\mathrm{eff}}(t-t')}\mathrm{e}^{-\mathcal{G}_{t'}}, \tag{13}$$

where $\mathcal{L}_{\mathrm{eff}}$ and $\mathcal{G}_t$ $(=\mathcal{G}_{t+T})$ are superoperators corresponding to the effective Liouvillian and the micromotion. Introducing the series expansion

$$\mathcal{L}_{\mathrm{eff}} = \sum_{k=0}^{\infty} \mathcal{L}_{\mathrm{eff}}^{(k)}; \qquad \mathcal{G}_t = \sum_{k=1}^{\infty} \mathcal{G}_t^{(k)}, \tag{14}$$

with $\mathcal{L}_{\mathrm{eff}}^{(k)} = O(\omega^{-k})$ and $\mathcal{G}_t^{(k)} = O(\omega^{-k})$, we obtain

$$\mathrm{i}\mathcal{L}_{\mathrm{eff}}^{(0)} = \mathrm{i}\mathcal{L}_0 \tag{15}$$

$$\mathrm{i}\mathcal{L}_{\mathrm{eff}}^{(1)} = \sum_{m\neq 0} \frac{[\mathrm{i}\mathcal{L}_{-m}, \mathrm{i}\mathcal{L}_m]}{2m\omega} \tag{16}$$

$$\mathrm{i}\mathcal{L}_{\mathrm{eff}}^{(2)} = \sum_{m\neq 0} \frac{[[\mathrm{i}\mathcal{L}_{-m}, \mathrm{i}\mathcal{L}_0], \mathrm{i}\mathcal{L}_m]}{2m^2\omega^2} + \sum_{m\neq 0}\sum_{n\neq 0,m} \frac{[[\mathrm{i}\mathcal{L}_{-m}, \mathrm{i}\mathcal{L}_{m-n}], \mathrm{i}\mathcal{L}_n]}{3mn\omega^2} \tag{17}$$

and

$$-\mathcal{G}_t^{(1)} = -\sum_{m\neq 0} \frac{\mathrm{i}\mathcal{L}_m}{m\omega} e^{-\mathrm{i}m\omega t} \tag{18}$$

$$-\mathcal{G}_t^{(2)} = \sum_{m\neq 0}\sum_{n\neq 0,m} \frac{[\mathrm{i}\mathcal{L}_n, \mathrm{i}\mathcal{L}_{m-n}]}{2mn\omega^2} e^{-\mathrm{i}m\omega t} + \sum_{m\neq 0} \frac{[\mathrm{i}\mathcal{L}_m, \mathrm{i}\mathcal{L}_0]}{m^2\omega^2} e^{-\mathrm{i}m\omega t}, \tag{19}$$

up to the second order of $1/\omega$. Here, $\mathcal{L}_m$ denotes the Fourier components of the Liouvillian:

$$\mathcal{L}_t = \sum_m \mathcal{L}_m \mathrm{e}^{-\mathrm{i}m\omega t}, \tag{20}$$

and the results have been obtained by the replacements

$$H_m \to \mathrm{i}\mathcal{L}_m, \tag{21}$$
$$H_{\mathrm{eff}} \to \mathrm{i}\mathcal{L}_{\mathrm{eff}}, \tag{22}$$
$$K(t) \to \mathrm{i}\mathcal{G}_t. \tag{23}$$

in the HF expansion for isolated systems (see Appendix A). One can obtain higher-order terms by continuing the procedure systematically.

We remark the relation to the Floquet–Magnus (FM) approach in the literature [50,63]. In this approach, we take a reference time $t = t_0$ and define the one-cycle evolution superoperator $\mathcal{V}_F(t_0) \equiv \mathcal{V}(t_0 + T, t_0)$. The Floquet Liouvillian $\mathcal{L}_F(t_0)$ is defined, in the FM approach, by

$$\mathcal{V}_F(t_0) = \mathrm{e}^{T\mathcal{L}_F(t_0)}. \tag{24}$$

On the other hand, in the van Vleck HF expansion [Eq. (13)], this one-cycle propagator is represented as $\mathcal{V}_F(t_0) = \mathrm{e}^{\mathcal{G}_{t_0}}\mathrm{e}^{T\mathcal{L}_{\mathrm{eff}}}\mathrm{e}^{-\mathcal{G}_{t_0}}$. By equating these, we have

$$\mathrm{e}^{T\mathcal{L}_F(t_0)} = \mathrm{e}^{\mathcal{G}_{t_0}}\mathrm{e}^{T\mathcal{L}_{\mathrm{eff}}}\mathrm{e}^{-\mathcal{G}_{t_0}}, \tag{25}$$

where we have used the periodicity $\mathcal{G}_{t_0+T} = \mathcal{G}_{t_0}$. Thus, the effective Liouvillians $\mathcal{L}_F(t_0)$ and $\mathcal{L}_{\mathrm{eff}}$ in the different approaches are related to each other by the similarity transformation $\mathrm{e}^{\mathcal{G}_{t_0}}$. The FM expansion is the series expansion for $\mathcal{L}_F(t_0)$ with $1/\omega$, and we have $\mathcal{L}_F(t_0) = \mathcal{L}_{\mathrm{eff}} = \mathcal{L}_0$ at the zeroth order of $1/\omega$. At higher orders, the $\mathcal{L}_F(t_0)$ involves more terms than $\mathcal{L}_{\mathrm{eff}}$ and the dependence on a reference time $t_0$ similarly to the case of isolated systems [9]. We note that $\mathcal{L}_F(t_0)$ contains the micromotion information.

Despite the formal analogy in the derivations of the van Vleck HF expansion in open and isolated systems, we need to be careful about the properties of $\mathcal{G}_t$ and $\mathcal{L}_{\mathrm{eff}}$. First, the micromotion superoperator $\mathrm{e}^{\mathcal{G}_t}$ is not unitary in general unlike $\mathrm{e}^{\mathrm{i}K(t)}$. Second, $\mathcal{L}_{\mathrm{eff}}$ may not be of Lindblad form. As shown in Refs. [64,65], even if the one-cycle evolution $\mathrm{e}^{T\mathcal{L}_F(t_0)}$ is a CPTP map, its logarithm $T\mathcal{L}_F(t_0)$ may not be a Lindbladian. One representative situation is when the one-cycle evolution $\mathrm{e}^{T\mathcal{L}_F(t_0)}$ has a negative real eigenvalue [64,65]. Noting that Eq. (25) implies that $\mathrm{e}^{T\mathcal{L}_F(t_0)}$ and $\mathrm{e}^{T\mathcal{L}_{\mathrm{eff}}}$ have the same eigenvalues[2], it also happens that $\mathcal{L}_{\mathrm{eff}}$ is not a Lindbladian. In addition, Mizuta et al. have recently shown that the FM expansion for $\mathcal{L}_F(t_0)$ may not be a Lindbladian [67]. In the following, we will show that, in the van Vleck approach, $\mathcal{L}_{\mathrm{eff}}$ is of Lindblad form at the first order of $1/\omega$ for physically relevant models. Another recent work [68] has also pointed out that $\mathcal{L}_{\mathrm{eff}}$ can be of Lindblad form while $\mathcal{L}_F(t_0)$ is not.

Nevertheless, $\mathcal{L}_{\mathrm{eff}}$ has a good property for analyzing the dynamics and NESSs: $\mathcal{L}_{\mathrm{eff}}$ has at least one eigenvalue equal to zero at every order of the HF expansion. To show this, we first prove the following lemma.

---

[2]Similarity transformations do not change eigenvalues [66]. Equation (25) means that $\mathrm{e}^{T\mathcal{L}_F(t_0)}$ and $\mathrm{e}^{T\mathcal{L}_{\mathrm{eff}}}$ are connected by a similarity transformation $\mathrm{e}^{\mathcal{G}_{t_0}}$.

**Lemma 1** *At each order of the HF expansion, we have*

$$\mathrm{tr}[\mathcal{L}_{\mathrm{eff}}(A)] = 0 \quad \text{for any operator } A. \tag{26}$$

**Proof.** We begin by proving

$$\mathrm{tr}[\mathcal{L}_m(A)] = \mathrm{tr}[\mathcal{H}_m(A) + \mathcal{D}_m(A)] = 0 \qquad \forall A \text{ and } \forall m. \tag{27}$$

To show Eq. (27), we first notice Eq. (6) implies $\mathrm{tr}[\mathcal{H}_m(A)] = -\mathrm{i}\,\mathrm{tr}([H_m, A]) = 0$, where we have used the cyclic property of the trace $\mathrm{tr}(H_m A) = \mathrm{tr}(A H_m)$. We second notice Eq. (8) leads to $\mathrm{tr}[\mathcal{D}(A)] = \mathrm{tr}[L_{\alpha,m-n} A L_{\alpha,n}^\dagger - \frac{1}{2}\{L_{\alpha,n}^\dagger L_{\alpha,m-n}, A\}] = \mathrm{tr}[L_{\alpha,n}^\dagger L_{\alpha,m-n} A - L_{\alpha,n}^\dagger L_{\alpha,m-n} A] = 0$, where we have again used the cyclic property. These two steps prove Eq. (27).

Now we prove Eq. (26) using Eq. (27) at each order $N$ of the HF expansion. As $\mathcal{L}_{\mathrm{eff}} = \sum_{k=0}^N \mathcal{L}_{\mathrm{eff}}^{(k)}$ and hence $\mathrm{tr}[\mathcal{L}_{\mathrm{eff}}(A)] = \sum_{k=0}^N \mathrm{tr}[\mathcal{L}_{\mathrm{eff}}^{(k)}(A)]$ at an $N$-th order, it is sufficient to prove

$$\mathrm{tr}[\mathcal{L}_{\mathrm{eff}}^{(k)}(A)] = 0 \qquad \forall A \text{ and } \forall k. \tag{28}$$

For $k = 0$, Eq. (28) is obtained from $\mathrm{tr}[\mathcal{L}_{\mathrm{eff}}^{(0)}(A)] = \mathrm{tr}[\mathcal{L}_0(A)] = 0$, where the last equality follows from Eq. (27) for $m = 0$. Let us hence prove Eq. (28) for $k \geq 1$. As represented in Eqs. (16) and (17), $\mathcal{L}_{\mathrm{eff}}^{(m)}$ consists of nested commutators between $\mathcal{L}_m$'s for different $m$'s. Unraveling all the commutators, we have the following formal expression $\mathcal{L}_{\mathrm{eff}}^{(k)} = \sum_{m_1,m_2,\ldots,m_k} C_{m_1,m_2,\ldots,m_k} \mathcal{L}_{m_1} \mathcal{L}_{m_2} \ldots \mathcal{L}_{m_k}$, where $C_{m_1,m_2,\ldots,m_k}$ are complex numbers. This expression gives

$$\mathrm{tr}[\mathcal{L}_{\mathrm{eff}}^{(k)}(A)] = \sum_{m_1,m_2,\ldots,m_k} C_{m_1,m_2,\ldots,m_k} \mathrm{tr}[(\mathcal{L}_{m_1}\mathcal{L}_{m_2}\ldots\mathcal{L}_{m_k})(A)] \tag{29}$$

$$= \sum_{m_1,m_2,\ldots,m_k} C_{m_1,m_2,\ldots,m_k} \mathrm{tr}[\mathcal{L}_{m_1}(A_{m_2,m_3,\ldots,m_k})], \tag{30}$$

where we have defined operators $A_{m_2,m_3,\ldots,m_k} \equiv (\mathcal{L}_{m_2}\ldots\mathcal{L}_{m_k})(A)$. Here, substituting $m = m_1$ and $A = A_{m_2,m_3,\ldots,m_k}$ into Eq. (27), we have $\mathrm{tr}[\mathcal{L}_{m_1}(A_{m_2,m_3,\ldots,m_k})] = 0$ holds for every set $(m_1, m_2, \ldots, m_k)$, meaning that $\mathrm{tr}[\mathcal{L}_{\mathrm{eff}}^{(k)}(A)] = 0$. Thus, we have proved Eq. (28) and hence Eq. (26). ∎

From this lemma, the following theorem holds true.

**Theorem 1** $\mathcal{L}_{\mathrm{eff}}$ *has at least one eigenvalue equal to zero at every order of the HF expansion.*

**Proof.** Recall that $\langle X, Y \rangle \equiv \mathrm{tr}(X^\dagger Y)$ serves as an inner product for two operators $X$ and $Y$. We translate Eq. (26) as $\langle I, \mathcal{L}_{\mathrm{eff}}(A) \rangle = 0 = \langle \mathcal{L}_{\mathrm{eff}}^*(I), A \rangle$, where $I$ is the identity operator and $\mathcal{L}_{\mathrm{eff}}^*$ is the adjoint superoperator[3] for $\mathcal{L}_{\mathrm{eff}}$. Remembering that $A$ can be any, we have $\mathcal{L}_{\mathrm{eff}}^*(I) = 0$, which means that $\mathcal{L}_{\mathrm{eff}}^*$ has a zero eigenvalue (and $I$ is the left eigenvector of $\mathcal{L}_{\mathrm{eff}}$). Therefore, $\mathcal{L}_{\mathrm{eff}}$ also has a zero eigenvalue ∎

As we will see below, the zero eigenvalue corresponds to the NESS.

In this work, we assume that $\mathcal{L}_{\mathrm{eff}}$'s eigenvalues all have nonpositive real parts. Since we have shown the existence of the zero eigenvalue, it means that the maximum of the eigenvalue real parts is zero. This property is important to obtain sensible time evolution since if $\mathcal{L}_{\mathrm{eff}}$ had

---

[3]For clarity, we use $A^\dagger$ for the adjoint operator for an operator $A$ and $\mathcal{L}^*$ for that for a superoperator $\mathcal{L}$.

an eigenvalue with a positive real part (and hence $\mathcal{V}_F$ had that with absolute value greater than 1), the density operator would blow up in many cycles of evolution. In physically relevant setups discussed in Secs. 5 and 6, we will see that $\mathcal{L}_{\text{eff}}$ is of Lindblad form at $O(\omega^{-1})$ and indeed has the good property. Thus, if the zero eigenvalue is not degenerate (as is the case in all example models in this paper), the higher-order corrections do not break the property for large enough $\omega$. If it is degenerate, there may appear small positive real parts at higher orders, and thus one must be careful about this possibility in general. We leave the general proof of the nonpositivity as an open problem and assume this throughout this work.

## 4    General aspects of NESS solution

With the van Vleck HF expansion discussed in the previous section, we can obtain the NESS solution, including the micromotion, without explicitly calculating time evolution. We discuss how it generally works in this section and will analyze physically relevant examples in the following sections.

Let us suppose that we have an arbitrary initial state $\rho_0$ at time $t = 0$ and consider its long-time evolution. Equations (9) and (13) give us the quantum state at time $t$ as

$$\rho(t) = e^{\mathcal{G}_t} e^{t\mathcal{L}_{\text{eff}}} e^{-\mathcal{G}_0} \rho_0. \tag{31}$$

In analyzing the long-time behavior, it is convenient to absorb the micromotion by the similarity transformation:

$$\rho'(t) \equiv e^{-\mathcal{G}_t} \rho(t), \tag{32}$$

which satisfies

$$\rho'(t) = e^{t\mathcal{L}_{\text{eff}}} \rho'(0). \tag{33}$$

In other words, $\rho'(t)$ obeys the time-independent master equation $\frac{d\rho'(t)}{dt} = \mathcal{L}_{\text{eff}}\rho'(t)$ with the initial condition $\rho'(t = 0) = e^{-\mathcal{G}_0}\rho_0$.

To investigate the long-time behavior of $\rho'(t)$, it is useful to introduce the eigenstates of $\mathcal{L}_{\text{eff}}$. For simplicity, we assume that $\mathcal{L}_{\text{eff}}$ is diagonalizable and introduce the eigenstates of $\mathcal{L}_{\text{eff}}$ as follows:

$$\mathcal{L}_{\text{eff}}(\eta_{\lambda,a}) = \lambda\eta_{\lambda,a}. \tag{34}$$

Here, $\eta_{\lambda,a}$ is the eigenstate of $\mathcal{L}_{\text{eff}}$ ($\eta_{\lambda,a}$ is a complex matrix acting on the Hilbert space) belonging to the complex eigenvalue $\lambda$, where $a$ ($= 1, 2, \ldots, N_\lambda$) labels the degenerate eigenstates with $N_\lambda$ being the degree of degeneracy. As discussed in the previous section, there exists $\lambda = 0$ eigenvalue, and we assume that $\text{Re}\lambda \leq 0$ for all $\lambda$.

We note that the trace preserving nature of the effective evolution $e^{t\mathcal{L}_{\text{eff}}}$ imposes the following condition,

$$\text{tr}(\eta_{\lambda,a}) = 0 \quad (\text{for } \lambda \neq 0). \tag{35}$$

Here the trace preserving nature means $\text{tr}\left(e^{t\mathcal{L}_{\text{eff}}}A\right) = \text{tr}(A)$ for any matrix $A$, which follows from $\text{tr}[\mathcal{L}_{\text{eff}}(A)] = 0$ at each order of the HF expansion. To prove Eq. (35), one considers $\rho'(t)$

starting from an arbitrary initial states $\rho'(0)$, which can be represented as a linear combination of $\eta_{\lambda,a}$'s (see Eqs. (37) and (38) below). Since $\text{tr}[\rho'(t)]$ is time-independent, one obtains that $\text{tr}(\eta_{\lambda,a})$ must vanish except for the zero modes $\lambda = 0$. For the zero modes, $\text{tr}(\eta_{0,a})$ can be nonvanishing, and, if so, we impose the following normalization

$$\text{tr}(\eta_{0,a}) = 1. \tag{36}$$

By using the eigenstates $\eta_{\lambda,a}$, we solve the asymptotic behavior of $\rho'(t)$. To this end, we express the initial state by these eigenstates as

$$\rho'(t=0) = \text{e}^{-\mathcal{G}_0}\rho_0 = \sum_{\lambda,a} c_{\lambda,a}\eta_{\lambda,a}, \tag{37}$$

which leads with Eq. (34) to

$$\rho'(t) = \sum_{\lambda,a} c_{\lambda,a}\text{e}^{\lambda t}\eta_{\lambda,a}. \tag{38}$$

Since $\text{Re}\lambda < 0$ holds true for all the $\lambda$'s except $\lambda = 0$, these eigenmodes all vanish in the long-time limit $t \to \infty$, and we have

$$\rho'(t) \to \rho'_\infty = \sum_a c_{0,a}\eta_{0,a}. \tag{39}$$

Thus, $\rho'(t)$ approaches a time-independent asymptotic state, which can be calculated with linear algebras for $\mathcal{L}_{\text{eff}}$ instead of explicit time-evolution simulations.

Once we have $\rho'_\infty$, we immediately obtain the NESS solution for $\rho(t)$,

$$\rho(t) \to \rho_{\text{ness}}(t) = \text{e}^{\mathcal{G}_t}\rho'_\infty = \sum_a c_{0,a}\text{e}^{\mathcal{G}_t}\eta_{0,a}. \tag{40}$$

The periodicity of $\mathcal{G}_t = \mathcal{G}_{t+T}$ ensures that the $\rho_{\text{ness}}$ is also periodic

$$\rho_{\text{ness}}(t+T) = \rho_{\text{ness}}(t). \tag{41}$$

Depending on whether $N_{\lambda=0} = 1$ or $N_{\lambda=0} \geq 2$, Eq. (40) gives different physical consequences. When $N_{\lambda=0} = 1$, the sum over $a$ is absent and we have $c_{0,1} = 1$ owing to the trace condition $\text{tr}(\rho'_\infty) = 1$. Thus, Eq. (40) is further simplified as

$$\rho_{\text{ness}}(t) = \text{e}^{\mathcal{G}_t}\eta_{0,1} \qquad (\text{when } N_{\lambda=0} = 1). \tag{42}$$

We emphasize that there is no dependence on the initial state $\rho_0$ in Eq. (42). When $N_{\lambda=0} = 1$, there is the unique steady state for $\mathcal{L}_{\text{eff}}$, and $\rho'(t)$ converges to this special state no matter what initial state we take. On the contrary, when $N_{\lambda=0} \geq 2$, there remain some initial state dependence in $c_{0,a}$'s. To obtain the NESS, we need to solve the linear equations [Eq. (37)] for the unknown coefficients $c_{\lambda,a}$, plugging $c_{\lambda=0,a}$ into Eq. (40). The multiple zero modes happen in physical models typically when the model has symmetries (see, e.g., Ref. [60]).

For completeness, we discuss how to calculate expectation values of observables in the NESS assuming Eq. (42). Using Eq. (14) together with Eqs. (18) and (19), we have $\rho_{\text{ness}}(t) = (1 + \mathcal{G}_t^{(1)} + \frac{[\mathcal{G}_t^{(1)}]^2}{2} + \mathcal{G}^{(2)}\ldots)\eta_{0,1} = \sum_m \rho_m e^{-im\omega t}$, where $\rho_m$ is time-independent and obtained up to a desired order of $\omega^{-1}$. For an observable $A$, we have its expectation value as $A(t) =$

$\text{tr}[\rho_{\text{ness}}(t)A] = \sum_m \text{tr}(\rho_m A)e^{-im\omega t}$. Therefore, once we have $\rho_m$'s from the HF expansion, we can immediately obtain expectation values at arbitrary times $t$ without time integration.

The above argument in deriving the NESS solution is exact once $\mathcal{L}_{\text{eff}}$ and $\mathcal{G}_t$ are given at an arbitrary order of the HF expansion. In this approach, the analytical expression for the approximate $\mathcal{L}_{\text{eff}}$ and $\mathcal{G}_t$ gives us physical intuitions for the effects of the drive and dissipation. Also, the HF expansion avoids dynamics simulations and enables efficient NESS calculations and possibly analytical calculations in some problems. For instance, one can use the Lanczos algorithm to numerically obtain $\eta_{0,1}$ more efficiently than the direct dynamics simulations. In the following sections, we classify the problems and develop and demonstrate the NESS calculations based on the HF expansion.

# 5  Phenomenological time-independent dissipators

In this section, we focus on the case in which the dissipator is time-independent $\mathcal{D}_t = \mathcal{D}$. This class of problems is widely studied in, e.g., quantum-optic [69], Rydberg atoms [70], cavity-QED [71], electronic [72,73], and spin [74–77] systems. Bloch equations used in magnets [54] and semiconductors [78,79] can also be viewed as equations of motion with time-independent dissipators. First, we reduce the HF expansion for the Liouvillian in Sec. 3 with the Hamiltonian and dissipator. Then, we apply the reduced formulas to the example model of a dissipative spin chain [80–82] and discuss what can be known by the HF expansion approach. We note that the van Vleck HF expansion for a time-independent dissipator was studied in Ref. [57] at the leading order $O(\omega^{-1})$. Here we study general dissipators at higher orders.

## 5.1  High-frequency expansion

When $\mathcal{D}_t$ is time-independent as $\mathcal{D}_t = \mathcal{D}$, we have $\mathcal{D}_m = \delta_{m0}\mathcal{D}$. Therefore, the Liouvillian Fourier components are simplified as

$$\mathcal{L}_m = \mathcal{H}_m + \delta_{m0}\mathcal{D}. \tag{43}$$

We substitute this special form of $\mathcal{L}_m$ into the Liouvillian HF expansion derived in Sec. 3, obtaining more concrete formulas.

Since the derivation is rather straightforward, we write down the results as follows:

$$\mathcal{L}_{\text{eff}}^{(0)}(\rho) = \mathcal{H}_0(\rho) + \mathcal{D}(\rho), \tag{44}$$

$$i\mathcal{L}_{\text{eff}}^{(1)}(\rho) = [H_{\text{eff}}^{(1)}, \rho], \tag{45}$$

$$i\mathcal{L}_{\text{eff}}^{(2)}(\rho) = [H_{\text{eff}}^{(2)}, \rho] + \sum_{m\neq 0} \frac{[[i\mathcal{H}_{-m}, i\mathcal{D}], i\mathcal{H}_m](\rho)}{2m^2\omega^2}, \tag{46}$$

and

$$-\mathcal{G}_t^{(1)}(\rho) = [iK^{(1)}(t), \rho], \tag{47}$$

$$-\mathcal{G}_t^{(2)}(\rho) = [iK^{(2)}(t), \rho] + \sum_{m\neq 0} \frac{[i\mathcal{H}_m, i\mathcal{D}](\rho)}{m^2\omega^2}e^{-im\omega t}. \tag{48}$$

Here, $H_{\text{eff}}^{(k)}$ and $K^{(k)}(t)$ are the effective Hamiltonian and micromotion defined in the HF expansion for isolated systems (see Appendix A for their explicit forms). In the derivation,

we have used a useful formula for nested commutators presented in Appendix B. Higher-order results are similarly obtained by straightforward calculations.

We remark that, up to the first order, the effective Liouvillian is in Lindblad form:

$$\mathcal{L}_{\text{eff}}(\rho) = -\mathrm{i}[H_{\text{eff}}, \rho] + \mathcal{D}(\rho) + O(\omega^{-2}), \tag{49}$$

which follows from Eqs. (44) and (45). This is a good property particular to the van Vleck approach and does not hold for $\mathcal{L}_F(t_0)$ [Eq. (24)] in the FM representation [67]. However, at higher orders, $\mathcal{L}_{\text{eff}}$ is not necessarily of Lindblad form due to the second term of the right-hand side in Eq. (46). These kinds of terms derive from the interplay between the external drive $H_m$ and dissipation $\mathcal{D}$. As for $\mathcal{G}_t$, Eq. (47) dictates that the micromotion is essentially the same as in isolated systems up to the first order, whereas the interplay sets in at the second order, as one can see in Eq. (48).

## 5.2 Example 1: Open XY Chain with Boundary Dissipation

Let us apply the HF expansion to an open XY spin chain subject to dissipation acting on the two edges of the chain. This class of models is mapped to quadratic Majorana fermions [80] and studied extensively. Here we discuss such a chain under periodic drive $f(t) = f(t+T)$ [83]:

$$H(t) = \sum_{j=1}^{N-1} \left( \frac{1+g}{2} \sigma_j^x \sigma_{j+1}^x + \frac{1-g}{2} \sigma_j^y \sigma_{j+1}^y \right) + h f(t) \sum_{j=1}^{n} \sigma_j^z \tag{50}$$

and the dissipator

$$\mathcal{D}(\rho) = \sum_{\alpha=1}^{4} \left( L_\alpha \rho L_\alpha^\dagger - \frac{1}{2} \left\{ L_\alpha^\dagger L_\alpha, \rho \right\} \right) \tag{51}$$

with

$$L_{1,2} = \sqrt{\Gamma_{1,2}^{\text{L}}} \sigma_1^\pm, \qquad L_{3,4} = \sqrt{\Gamma_{1,2}^{\text{R}}} \sigma_N^\pm. \tag{52}$$

In the literature, this model was first studied analytically for the periodic kicks $f(t) = \delta_T(t) \equiv T \sum_{m \in \mathbb{Z}} \delta(t - mT)$ [83]. The authors showed the following two properties:

(i) This model exhibits a rich phase diagram in the $(T, h)$-plane,

(ii) the boundary dissipations $\{L_\alpha\}_{\alpha=1}^4$ do not qualitatively change the phase diagram.

Here, the rich phase diagram was characterized by the spin-spin correlations [83], and, its correspondence to the topological edge mode was elucidated afterwords [81,82]. Our aim here is to reexamine this model by applying our HF expansion method and to provide further insights. Note that the HF expansion approach can be used for various driving protocols $f(t)$ other than $\delta_T(t)$ in exchange for restricting ourselves to the HF regime.

We begin by considering the property (i) described above by the HF expansion. As noted above, this property is related to the bulk Hamiltonian (the bulk-boundary correspondence was confirmed for the topological edge mode [81]). Thus, we neglect the dissipator and consider the infinite chain in discussing the property (i).

The spin Hamiltonian (50) is many-body and hard to solve analytically, and it is convenient to map this to Majorana fermions, or the two Hermitian components of the complex Jordan-Wigner fermions. We introduce $2N$ Majorana fermions $w_j$ ($j = 1, 2, \ldots, 2N$) for an $N$-site chain as

$$w_{2n-1} = \left( \prod_{j=1}^{n-1} \sigma_j^z \right) \sigma_n^x, \qquad w_{2n} = \left( \prod_{j=1}^{n-1} \sigma_j^z \right) \sigma_n^y, \qquad w_{2n-1} w_{2n} = i\sigma_n^z, \tag{53}$$

which satisfy the Majorana commutation relations $\{w_i, w_j\} = 2\delta_{ij}$ and translate Eq. (50) into the following quadratic Hamiltonian

$$H(t) = -i \sum_{j=1}^{N-1} \left( \frac{1+\gamma}{2} w_{2j} w_{2j+1} - \frac{1-\gamma}{2} w_{2j-1} w_{2(j+1)} \right) - ihf(t) \sum_{j=1}^{N} w_{2j-1} w_{2j}. \tag{54}$$

The Majorana fermions are interpreted as spinon excitations behaving as magnetic domain walls. From the Majorana fermions to the spins, we can use the inverse transformation of Eq. (53) given by

$$\sigma_n^x = (-i)^{n-1} \left( \prod_{j=1}^{2(n-1)} w_j \right) w_{2n-1}, \quad \sigma_n^y = (-i)^{n-1} \left( \prod_{j=1}^{2(n-1)} w_j \right) w_{2n}, \quad \sigma_n^z = -iw_{2n-1} w_{2n}. \tag{55}$$

It is convenient to introduce the two-component fermions $W_j = {}^t(w_{2j-1}, w_{2j})$ and make the Fourier transform $W_j \to \widetilde{W}_k \propto \sum_j e^{-ikj} W_j$ in the limit of $N \to \infty$. Then we have the Heisenberg equation for $\widetilde{W}_k$ (see Appendix C for derivation):

$$\frac{d\widetilde{W}_k(t)}{dt} = -ih(k, t)\widetilde{W}_k(t), \tag{56}$$

where

$$h(k, t) = -2\{(g\sin k)\tau^x + [\cos k - hf(t)]\tau^y\}, \tag{57}$$

where $\tau^\alpha$ ($\alpha = x, y,$ and $z$) denote the Pauli matrices. Thus, the original spin Hamiltonian has been mapped to noninteracting two-component Majorana fermions. Although the mapping (53) is nonlinear and does not simply tell us every spin observable, the two eigenvalues of Eq. (57) describe the instantaneous energy dispersion relations for the elementary spinon (Majorana fermion) excitations.

As shown in Refs. [74, 83], the nontrivial phase is related to the quasienergy bands of the elementary Majorana fermions. Since $h(k, t)$ is Hermitian and traceless, the two eigenvalues of the one-cycle unitary $V(k) = \exp_+[-i \int_0^T H(k, t)dt]$ are given as $\exp[\pm i\epsilon(k)T]$, and $\pm\epsilon(k)$ determine the two quasienergy dispersion relation for the Floquet modes in $k \in (-\pi, \pi]$. The appearance of the nontrivial phase is signaled by the appearance of the nontrivial solutions for $d\epsilon(k)/dk = 0$ [74, 83]. Here, the nontrivial solutions mean $k \neq 0$ or $\pi$, since, without the periodic drive, or $f(t)=0$, the two quasienergy bands are given by $\pm\epsilon = \pm 2\sqrt{\cos^2 k + g^2 \sin^2 k}$ and $d\epsilon(k)/dk = 0$ at $k = 0$ and $\pi$.

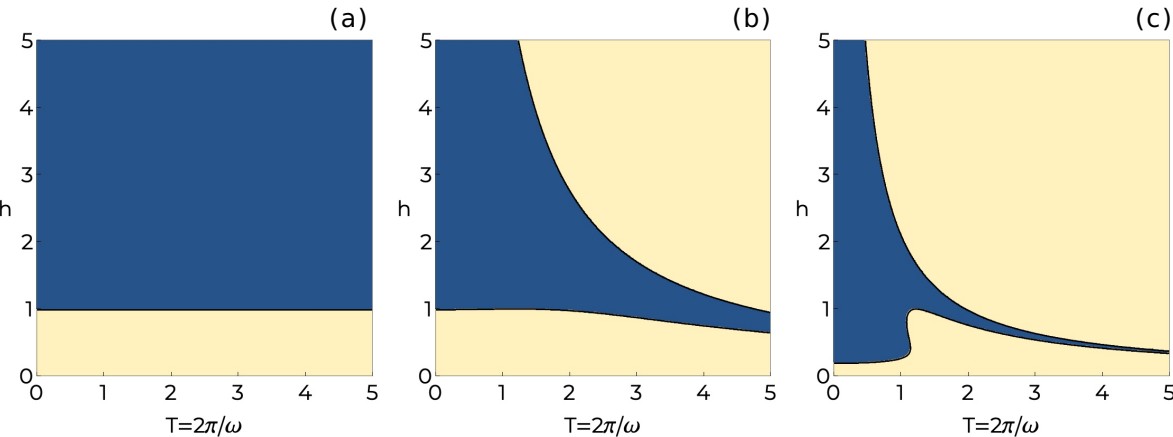

Figure 1: Number of solutions for $d\epsilon(k)/dk = 0$ in $k \in (0,\pi)$ obtained by the HF expansion. The blue (yellow) region represents the number 0 (1). The zeroth-order (a) and second-order (b) calculation for $g = 0.1$. (c) The second-order calculation for $g = 0.9$.

Let us now derive the phase diagram by applying the HF (or small $T$) expansion to Eq. (57) for the periodic kicks $f(t) = \delta_T(t)$. In this example, the Fourier components of the Hamiltonian $h_m(k) = \int_0^T dt\, h(k,t) e^{im\omega t}/T$ are simply given by $h_{m=0} = \overline{h(k,t)} = -2\{g \sin k \tau^x + (\cos k - h)\tau^y\}$ and $h_{m \neq 0} = 2h\tau^y$. At the zeroth order of the HF expansion, we have $h_{\text{eff}}^{(0)}(k) = h_{m=0}$, which gives the quasienergy bands $\pm\epsilon^{(0)}(k) = \pm 2\sqrt{(\cos k - h)^2 + g^2 \sin^2 k}$. In this order, $\epsilon^{(0)}(k)$ does not depend on $T$, and the number of solutions in $d\epsilon^{(0)}(k)/dk = 0$ changes whether $|h| \geq 1$ or $|h| < 1$. Whereas, for $|h| \geq 1$, the only solutions in $k \in [0,\pi]$ are $k = 0$ and $\pi$, there is an extra solution $k = k_* \in (0,\pi)$ for $|h| < 1$. Correspondingly, the spin-spin correlations [83] and the number of topological edge modes become different in these parameter regions as shown in Fig. 1(a). At the first order of the HF expansion, $h_{\text{eff}}$ does not acquire a correction: $h_{\text{eff}}^{(1)} = 0$. This follows from the time-reversal symmetry of the driving protocol, $f(-t) = f(t)$, and $h_{\text{eff}}^{(2n+1)} = 0$ ($n \in \mathbb{N}$) more generally. At the second order, we obtain a nonvanishing correction from $h_{\text{eff}}^{(2)}$, and the effective Hamiltonian up to this order is given by $h_{\text{eff}} = -2\{g[1 - 2(hT)^2/3]\sin k \tau^x + (\cos k - h)\tau^y\}$. We note that the periodic kick in the Zeeman coupling has reduced the anisotropy $|g|$. In this order, $h_{\text{eff}}$ depends on $T$ and gives more complex phase diagrams as shown in Figs. 1(b) and (c). We remark that the phase diagrams are consistent with the exact solution [83] at the small-$T$ (i.e., high-frequency) region. We leave it an open question to examine how the phase diagram changes for higher-order calculations, which require considerable effort.

One advantage of the HF expansion approach is that we can analyze other driving protocols rather than periodic kicks. While the problem is exactly solvable for the kicks, it is not for, e.g., the harmonic drive $hf(t) = b + a\cos\omega t$. To obtain the phase diagram for this drive, we decompose $h(k,t)$ into its Fourier components: $h_{m=0}$ is the same as before, $h_{m=\pm 1} = h\sigma_y$, and $h_m = 0$ otherwise. Thus, at the zeroth order, we obtain the same phase diagram as before (see Fig. 1(a)). Again, all the odd-order corrections vanish in the harmonic drive as well, and the first nontrivial correction comes at the second order. The effective Hamiltonian up to the second order is given by $h_{\text{eff}} = -2\{g[1 - (bT)^2/\pi^2]\sin k \tau^x + (\cos k - a)\tau^y\}$, in which we find a quantitative difference in the change of anisotropy $g$.

Now, we discuss the other property (ii). As we have seen in Sec. 5.1, the effective Liouvillian involves the contributions from the interplay between the drive and dissipation (see the second terms on the RHS of Eqs. (46) and (48)). These terms, in general, act on sites other than the two edges even if the bare dissipator $\mathcal{D}$ acts only on the edges. As we go higher orders, the dissipators propagate into the bulk and can cause a nonnegligible effect on the bulk property in general.

Nevertheless, we can show that the present model is so special that all drive-dissipation-interplay terms vanish and

$$\mathcal{L}_{\text{eff}}\rho = -\mathrm{i}[H_{\text{eff}}, \rho] + \mathcal{D}(\rho) \tag{58}$$

holds at each order of the HF expansion. Namely, the drive does not let the dissipator propagate into the bulk, and the effective dynamics is described by the effective Hamiltonian plus the bare boundary dissipation.

To prove Eq. (58), it is enough to show $[\mathcal{H}_m, \mathcal{D}] \propto [\mathcal{S}_z, \mathcal{D}] = 0$, where $\mathcal{S}_z$ denotes the superoperator $\mathcal{S}_z\rho \equiv [\sum_j \sigma_j^z, \rho]$. Note that $\mathcal{H}_m \propto \mathcal{S}_z$ for $m \neq 0$ because $H_m \propto \sum_j \sigma_j^z$ (see Eq. (50)). One can easily show this by straightforward calculations for the superoperator commutators, but this approach cannot gain physical insights. Here we provide another proof based on the vectorization of the operator space [66]. In this technique, the density matrix $\rho$ acting on the Hilbert space $\mathfrak{H}$ is regarded as a vector $\vec{\rho}$ in $\mathfrak{H} \otimes \mathfrak{H}$. Correspondingly, a superoperator $\rho \to L\rho R$ with $L$ and $R$ being matrices acting on $\mathfrak{H}$ is regarded as a matrix $L \otimes R^T$ ($T$ denotes the transpose) acting on $\mathfrak{H} \otimes \mathfrak{H}$. Graphically, each superoperator becomes an operator acting on two copies of spin chains, as illustrated in Fig. 2. In our model, the dissipator is represented as

$$\mathcal{D} = \mathcal{D}_1 + \mathcal{D}_2 \tag{59}$$

$$\mathcal{D}_1 \equiv \sum_{\alpha=1}^4 L_\alpha \otimes (L_\alpha^\dagger)^T = \Gamma_1^{\mathrm{L}}\sigma_1^+ \otimes \sigma_1^+ + \Gamma_2^{\mathrm{L}}\sigma_1^- \otimes \sigma_1^- + \Gamma_1^{\mathrm{R}}\sigma_N^+ \otimes \sigma_N^+ + \Gamma_2^{\mathrm{R}}\sigma_N^- \otimes \sigma_N^-, \tag{60}$$

$$\mathcal{D}_2 \equiv -\frac{1}{2}\sum_{\alpha=1}^4 \left[ L_\alpha^\dagger L_\alpha \otimes 1 + 1 \otimes (L_\alpha^\dagger L_\alpha)^T \right] = -\frac{1}{2}\left( K \otimes 1 + 1 \otimes K \right), \tag{61}$$

$$K \equiv \Gamma_1^{\mathrm{L}}\sigma_1^-\sigma_1^+ + \Gamma_2^{\mathrm{L}}\sigma_1^+\sigma_1^- + \Gamma_1^{\mathrm{R}}\sigma_N^-\sigma_N^+ + \Gamma_2^{\mathrm{R}}\sigma_N^+\sigma_N^-$$
$$= \frac{\Gamma_2^{\mathrm{L}} - \Gamma_1^{\mathrm{L}}}{2}\sigma_1^z + \frac{\Gamma_2^{\mathrm{R}} - \Gamma_1^{\mathrm{R}}}{2}\sigma_N^z + \frac{\Gamma_1^{\mathrm{L}} + \Gamma_2^{\mathrm{L}} + \Gamma_1^{\mathrm{R}} + \Gamma_2^{\mathrm{R}}}{2} \tag{62}$$

where we have used $\sigma_j^+\sigma_j^- = (1 + \sigma_j^z)/2$. Although the explicit form involves many terms, its physical meaning is obvious if we think of $\mathcal{D}$ as a virtual Hamiltonian acting on the two spin chains. First, $\mathcal{D}_1$ consists of pairwise spin raising or lowering on the left or right edges, as shown in Fig. 2(a). Second, $\mathcal{D}_2$ gives local Zeeman energy on each corner of the two chains, where $K \otimes 1$ ($1 \otimes K$) acts on the upper (lower) chain as illustrated in Fig. 2(b). Thus, $\mathcal{D}$ does not conserve the total spin along the $z$ direction but conserves the difference between the in-chain total spins along $z$. Now, we recall that $\mathcal{S}_z$ is given in the matrix representation as

$$\mathcal{S}_z = \sum_j \sigma_j^z \otimes 1 - 1 \otimes \sum_j \sigma_j^z, \tag{63}$$

which is the in-chain spin difference. Therefore, $\mathcal{S}_z$ is a conserved quantity under the "Hamiltonian" $\mathcal{D}$, and we obtain $[\mathcal{S}_z, \mathcal{D}] = 0$ that means $[\mathcal{H}_m, \mathcal{D}] = 0$ ($\forall m$) leading to Eq. (58).

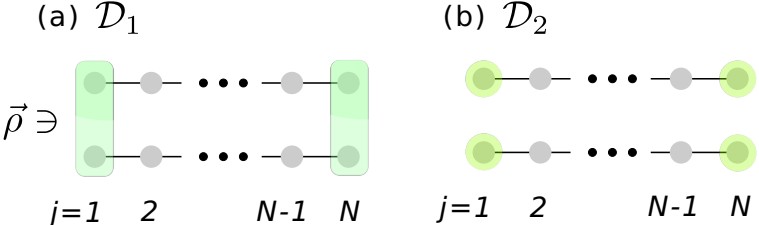

Figure 2: Schematic illustration of matrices (a) $\mathcal{D}_1$ and (b) $\mathcal{D}_2$ in the vectorization scheme of $\rho \to \vec{\rho} \in \mathfrak{H} \otimes \mathfrak{H}$. The upper (lower) chain corresponds to the left (right) part of $\mathfrak{H} \otimes \mathfrak{H}$. The colored sites show nontrivial actions of parts of $\mathcal{D}_1$ and $\mathcal{D}_2$ (see Eqs. (60)–(62)).

We remark that the disentangling of the drive and dissipation is particular to the present dissipator (52), which is absent in generic models (another special case is the dephasing $L_\alpha \propto \sigma_j^z$). For example, we can think of other dissipators such as spin raising and lowering along $x$ instead. In this case, $\mathcal{D}_1$ involves the terms like $\sigma_1^+ \otimes \sigma_1^-$, and the difference between the in-chain total $\sigma^z$ is no longer conserved, implying that the drive-dissipation-interplay could influence the bulk property more.

Let us summarize this subsection. We have studied the prominent dissipative open spin chain, addressing the issues (i) and (ii) from the HF expansion viewpoint. As for (i), we have provided a way to analyze generic driving protocol $f(t)$ at high frequency. As for (ii), we have found the drive-dissipation disentangling, which means that the effective Liouvillian is of Lindblad form, and the dissipator is confined in the spin chain edges.

### 5.3   Example 2: Three-level system

Here we demonstrate, more quantitatively, the NESS calculation by the HF expansion. To this end, we take an effective Hamiltonian for the NV center in diamonds under a circularly-polarized ac magnetic field [84]:

$$H_{\mathrm{NV}}(t) = H_0 + H_{\mathrm{ext}}^{\mathrm{circ}}(t), \tag{64}$$

$$H_0 = -B_s S_z + N_z S_z^2 + N_{xy}(S_x^2 - S_y^2), \qquad H_{\mathrm{ext}}^{\mathrm{circ}}(t) \equiv -B_d(S_x \cos\omega t + S_y \sin\omega t), \tag{65}$$

where $S_{x,y,z}$ are the spin-1 operators, $B_s$ is the static Zeeman field, $N_z$ and $N_{xy}$ represent the magnetic anisotropy terms, and $H_{\mathrm{ext}}^{\mathrm{circ}}(t)$ denotes the coupling to the circularly-polarized ac magnetic field. In our analysis, we set $N_z = 1$ and $N_{xy} = 0.05$ because of $N_z \gg N_{xy}$ in NV centers [84]. Since we consider the spin $S = 1$, the spin matrices are $3 \times 3$. This model was analyzed in Ref. [57] at $O(\omega^{-1})$, and we here extend the analysis to $O(\omega^{-2})$. We write the $H_0$'s eigenstates as $H_0 |E_i\rangle = E_i |E_i\rangle$ ($i = 1, 2,$ and 3).

As in Ref. [57], we here consider time-independent dissipation and will analyze time-dependent dissipation to compare the two cases in Sec. 6.2. To make the comparison easy, we use the dissipator that represents the coupling of $S_x$ to an ohmic bosonic thermal reservoir [40]:

$$\mathcal{D}(\rho) = \sum_{i,j} \gamma(E_{ij}) \left[ A_{E_{ij}} \rho A_{E_{ij}}^\dagger - \frac{1}{2} \left\{ A_{E_{ij}}^\dagger A_{E_{ij}}, \rho \right\} \right], \tag{66}$$

where $E_{ij} \equiv E_i - E_j$, $A_{E_{ij}} = \Pi_{E_i} S_x \Pi_{E_j}$ ($\Pi_{E_i} \equiv |E_i\rangle \langle E_i|$), and

$$\gamma(\epsilon) = \gamma_0 \frac{\epsilon e^{-\frac{\epsilon^2}{2\Lambda^2}}}{1 - e^{-\beta\epsilon}} \tag{67}$$

is the bath spectral function. Here, $\gamma_0$ gives the system-bath coupling strength, $\beta$ the inverse temperature, and $\Lambda$ the spectral cutoff. In the following numerical calculations, we set $\gamma_0 = 0.2$, $\beta = 3$, and $\Lambda = 10$.

We remark that the dissipator (66) is approximate because it is represented by the energy eigenvalues and eigenstates for the undriven $H_0$. Later in Sec. 6.2, we will analyze a more precise dissipator that is derived microscopically and involves the driving effect, finding that the dissipator difference only gives small quantitative corrections except some symmetry. Given that similar approximate time-independent dissipators are widely used in NV-center studies [84], the comparison between Secs. 5.3 and 6.2 offers a theoretical validation for the approximation.

The dissipator (66) satisfies the detailed balance condition. To see this, we rewrite Eq. (66) as

$$D(\rho) = \sum_{i,j} \Gamma_{ij} \left[ L_{ij} \rho L_{ij}^\dagger - \frac{1}{2} \left\{ L_{ij}^\dagger L_{ij}, \rho \right\} \right], \tag{68}$$

where $L_{ij} = |E_i\rangle \langle E_j|$ are jump operators between the energy eigenstates, and $\Gamma_{ij} = \gamma(E_{ij}) |\langle E_i|S_x|E_j\rangle|^2$ are the corresponding transition rates. Noting that $\gamma(-\epsilon) = e^{-\beta\epsilon}\gamma(\epsilon)$, we have $\Gamma_{ij}e^{-\beta E_j} = \Gamma_{ji}e^{-\beta E_i}$. This detailed balance condition ensures that, without drive $H_{\text{ext}}^{\text{circ}}(t)$, the stationary solution of the Lindblad equation corresponds to the canonical ensemble $\rho_{\text{can}} = e^{-\beta H_0}/Z$ with $Z = \text{tr}(e^{-\beta H_0})$.

In this setup, we calculate the NESS for the Lindblad equation in two ways. The first way is the numerical time-integration of the Lindblad equation $\frac{d\rho}{dt} = \mathcal{L}_t\rho$ for $\mathcal{L}_t\rho = -i[H_{\text{NV}}(t), \rho] + \mathcal{D}(\rho)$. By solving for independent initial conditions, we numerically obtain all the matrix elements for the one-cycle superoperator ($9 \times 9$ matrix) $\mathcal{V}_F$. Then we diagonalize $\mathcal{V}_F$, finding $\eta$ as the eigenvector with eigenvalue 1. Finally, the NESS $\rho_{\text{ness}}(t)$ is obtained by the time-integration starting from $\eta$. For clarity, we write $\rho_{\text{ness}}(t)$ thus obtained as $\rho_{\text{ness}}^{\text{exact}}(t)$. This way of calculation is exact but demands many time integrations.

The second way of calculation is the HF expansion approach. Following Eqs. (44)–(46), we write out $\mathcal{L}_{\text{eff}}$ at an order $N$ ($= 0, 1, 2, \dots$). Then we numerically find $\eta$ as the eigenvector of $\mathcal{L}_{\text{eff}}$ with zero eigenvalue. Then we obtain $\rho_{\text{ness}}^{(N)}(t) = e^{\mathcal{G}_t}\eta$, where $\mathcal{G}_t$ is calculated from Eqs. (47) and (48), depending on $N$. In this approach, we do not need any time-integration but only use linear algebra. At the first order [57], $\rho_{\text{ness}}^{(N=1)}(t)$ was exactly solved for any $\gamma_0$ and shown to coincides with the canonical Floquet steady state in the limit of $\gamma_0 \to 0$. We note that the correction terms to $H_{\text{eff}}$ in the present model are given by

$$H_{\text{eff}}^{(1)} = \frac{2B_d^2}{\omega} S_z, \qquad H_{\text{eff}}^{(2)} = -\frac{2B_d^2}{\omega^2} \left[ -B_s S_z + 3N_z S_z^2 + N_{xy}(S_x^2 - S_y^2) \right]. \tag{69}$$

Whereas the first term represents an effective Zeeman field, the second modifies the nematic

terms $N_z$ and $N_{xy}$. The corrections to the micromotion read

$$K^{(1)}(t) = -\frac{2B_d}{\omega}[\sin(\omega t)S_x + \cos(\omega t)S_y], \tag{70}$$

$$K^{(2)}(t) = -\frac{2B_d B_s}{\omega^2}[\sin(\omega t)S_x + \cos(\omega t)S_y]$$
$$+ \frac{2B_d(N_{xy} + N_z)}{\omega^2}\cos(\omega t)\{S_y, S_z\} - \frac{2B_d(N_{xy} - N_z)}{\omega^2}\cos(\omega t)\{S_z, S_x\}, \tag{71}$$

which consist of circularly polarized ac magnetic field and similar terms for nematics. The drive-dissipation-interplay terms are not written in simple form and hence treated just numerically.

The one-cycle evolutions for the NESS calculated with these two ways are plotted in Fig. 3, where we have set $B_s = 0.3$, $B_d = 0.1$, $\omega = 10$, $\beta = 3$, $\gamma_0 = 0.2$, and $\Lambda = 10$. For the HF approach, we plot the results for the zeroth, first, and second orders. At the zeroth order, $H_{\text{eff}} = H_0$ and $\mathcal{G}(t) = 0$, and thus the NESS corresponds to the thermal equilibrium showing no time dependence. As we increase the order, the HF result tends to approach the exact numerical integration, which we verify quantitatively below. We note that there are some period-$T/2$ (or the second harmonic) oscillations in, e.g., $S_x^2 - S_y^2$ in Fig. 3. These oscillations cannot be taken up to the first order since the micromotion $K^{(1)}(t)$ contains only frequency $\omega$. At the second order, $\rho_{\text{ness}}^{(N=2)}(t) = e^{\mathcal{G}_t}\eta = (1 + \mathcal{G}_t + \mathcal{G}_t^2)\eta$, where $\mathcal{G}_t^2\eta$ involves the frequency $2\omega$.

The HF expansion approach becomes more accurate for higher frequency $\omega$. Figure 4 show the $\omega$-dependence of one-cycle averages of observables

$$\overline{O} = \int_0^T \frac{dt}{T}\text{tr}[\rho_{\text{ness}}(t)O] \tag{72}$$

for $O = S_z, S_x^2 - S_y^2, S_z^2$, and $\{S_x, S_y\} = \frac{1}{2i}((S^+)^2 - (S^-)^2)$ in the NESS ($S^{\pm} = S_x \pm iS_y$). We note that the other observables have zero one-cycle averages for symmetry reasons, as shown in Ref. [57]. We observe that the higher-order HF expansion reproduces the exact results. To quantify the goodness of the HF expansion approach more strictly, we introduce the following measure

$$\delta\rho_N = \left[\int_0^T \frac{dt}{T}\|\rho_{\text{ness}}^{(N)}(t) - \rho_{\text{ness}}^{\text{exact}}(t)\|^2\right]^{1/2}, \tag{73}$$

where $\|\cdots\|$ denotes the Hilbert-Schmidt norm for matrices. If $\delta\rho_N = 0$, it follows that $\rho_{\text{ness}}^{(N)}(t) = \rho_{\text{ness}}^{\text{exact}}(t)$ for any $t$. Figure 5 shows $\delta\rho$ for the orders $0, 1$, and $2$ plotted against $\omega$. We observe

$$\delta\rho_N \propto \frac{1}{\omega^{N+1}} \tag{74}$$

for high frequencies $\omega \gtrsim 2$. This is a clear indication that the $N$-th order approximation $\rho_{\text{ness}}^{(N)}(t)$ completely describe $\rho_{\text{ness}}^{\text{exact}}(t)$ up to $O(\omega^{-N})$.

Let us summarize this subsection. We have used the HF expansion approach to calculate the NESS quantitatively. This approach does not require numerical time integration but uses linear algebra. In this approach, the effective Hamiltonian and micromotion are interpretable.

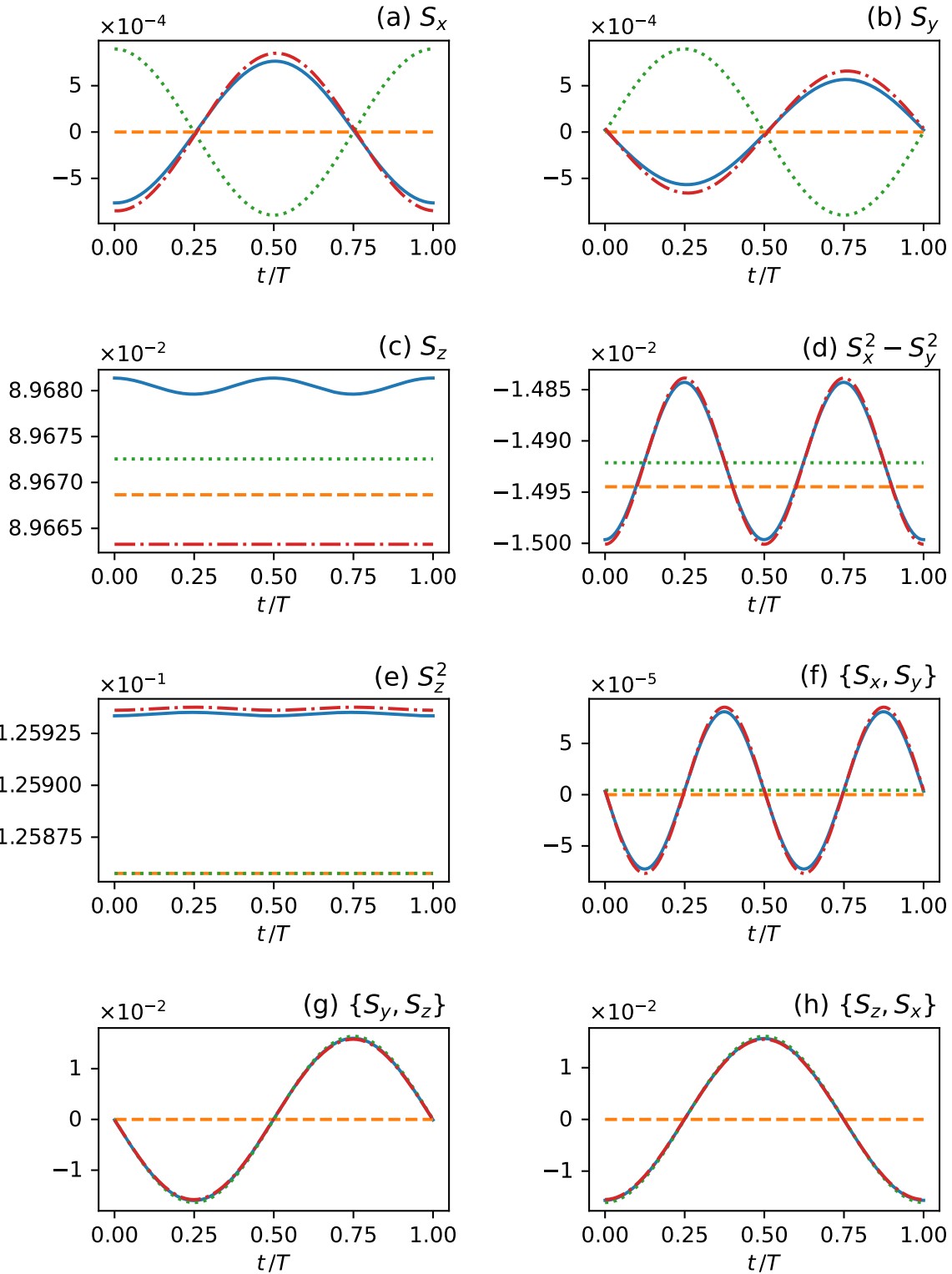

Figure 3: NESS in one cycle calculated by the exact numerical integration (blue) and by the HF expansion approach at order 0 (orange), 1 (green), and 2 (red). All the independent 8 observables are plotted in each panel. The parameters are chosen as $B_s = 0.3$, $N_z = 1$, $N_{xy} = 0.05$, $B_d = 0.1$, $\omega = 10$, $\beta = 3$, $\gamma_0 = 0.2$, and $\Lambda = 10$.

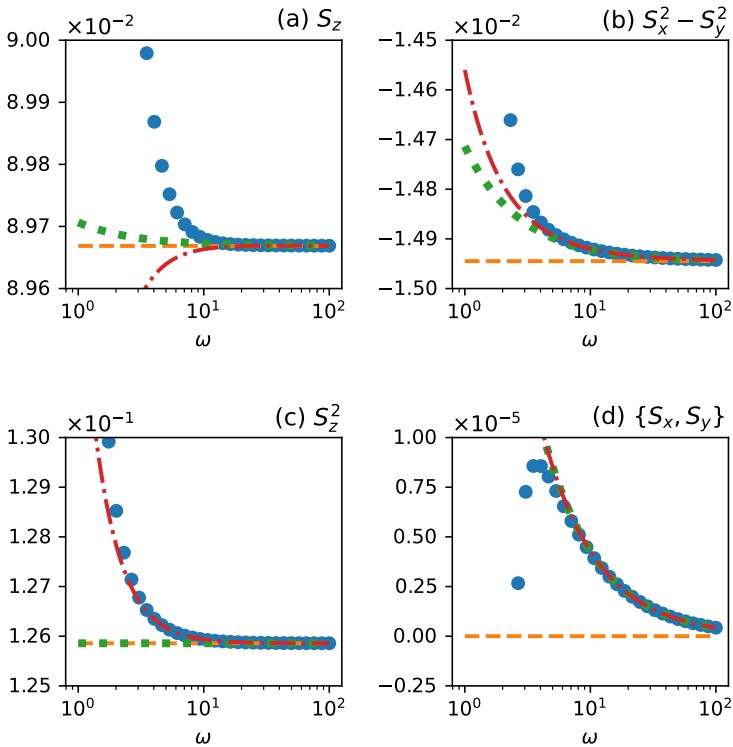

Figure 4: One-cycle averages of observables in the NESS [Eq. (72)] plotted against $\omega$. Here $\rho_{\mathrm{ness}}(t)$ is calculated by the exact numerical integration (blue) and by the HF expansion approach at order 0 (orange), 1 (green), and 2 (red). The parameters are the same as in Fig. 3 except $\omega$.

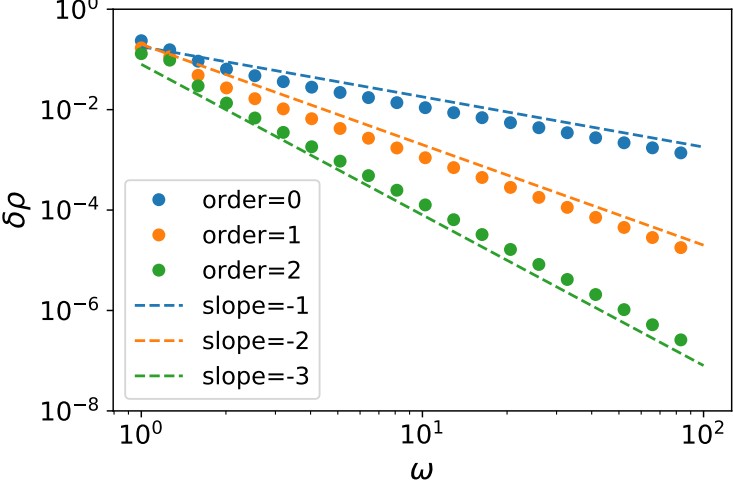

Figure 5: Difference between NESSs $\rho_{\mathrm{ness}}^{\mathrm{exact}}(t)$ and $\rho_{\mathrm{ness}}^{(N)}(t)$ [Eq. (73)] plotted against $\omega$. The data points are the results for $N = 0$ (blue), 1 (orange), and 2 (green), and the dashed lines are guides to the eye for some slopes (see legend).

Although the drive-dissipation-interplay terms are difficult to interpret, they are important to reproduce the exact results quantitatively.

As discussed at the end of Sec. 4, this approach is numerically efficient especially for many-body systems. To obtain the NESS, the time integration requires many matrix-vector multiplications (i.e., applications of $\mathcal{L}_t$ onto $\rho(t)$) before the NESS is reached. The number of multiplications is estimated by $\sim 1/(\gamma_0 \Delta t)$, where $\gamma_0$ is the dissipation strength and $\Delta t$ is the time stepping. To obtain high-accuracy results, we have to decrease $\Delta t$ and need numerous multiplications. On the other hand, in the HF-expansion approach, we are to find the eigenstate $\eta_{0,1}$ of $\mathcal{L}_{\text{eff}}$ with the largest real part. For this purpose, we can use the famous Lanczos algorithm, where the required number of matrix-vector multiplications are greatly suppressed. So we can reach the NESS more efficiently with the HF-expansion approach within the HF approximation. The efficiency also applies to calculating physical observables as discussed at the end of Sec. 4.

# 6 Microscopically-derived dissipators by weak thermal contact

In Sec. 5, we have discussed phenomenological Floquet-Lindblad equations, where the dissipators are time-independent. However, those dissipators are only approximate, strictly speaking. The dissipator originates from the fact that the system of interest is coupled weakly to its environment [40], and thus it can be modified and time-dependent when the system is driven periodically.

In this section, we discuss time-dependent dissipators that are microscopically derived for the system-bath coupling setup. The most well-known example is the quantum master equation obtained by the rotating wave approximation (RWA) [41, 85]. Also, there are other time-dependent Lindblad equations that have recently been derived without the RWA [86–88]. In these FLEs with or without the RWA, one can apply the HF expansion for the Liouvillian derived in Sec. 3 to the expansions for the Hamiltonian, Lamb shift, and dissipator. Unlike time-independent ones, time-dependent dissipators have multiple Fourier components, which induce more terms in the HF expansion. One subtlety is that the Lamb shift and dissipator cannot be Taylor-expanded for $\omega^{-1}$ in general because the bath spectral function $\gamma(\epsilon)$ cannot be Taylor-expanded (see, e.g., Eq. (67)) but depend on $\omega$ in it if the dissipator is affected by the drive. Nevertheless, one can still use the HF expansion if we treat these $\omega$-dependence rigorously while HF-expand other $\omega^{-1}$ dependences (see a related discussion below in Sec. 6.1.3).

Throughout this section, we focus on a widely-accepted special class of FLEs: the quantum master equation obtained by the rotating wave approximation (RWA) [41, 85]. As we will see below, these FLEs have the special property that the time-dependence can be eliminated by an appropriate unitary transformation. Thus, we will derive a different HF-expansion-based approach to analyze the NESS utilizing this useful special property. This new approach is different from the previous one derived in Secs. 3 and 4, but one could also use the previous one to the examples in this section.

## 6.1 Floquet-Lindblad equation obtained by RWA

### 6.1.1 Floquet-Lindblad equation and its characteristic properties

Leaving its derivation in Appendix D, we here summarize the FLE obtained microscopically with the RWA. Suppose that we have a system-bath coupled Hamiltonian, $H_{\text{tot}}(t) = H_S(t) + H_B + H_{SB}$, where $H_S(t+T) = H_S(t)$ is for the periodically driven system of interest, $H_B$ for the heat bath (reservoir), and $H_{SB} = \sum_\alpha A_\alpha \otimes B_\alpha$ for the system-bath coupling. The bath is characterized by the correlator,

$$\Gamma_{\alpha\beta}(\omega) = \int_0^\infty ds\, e^{i\omega s} \langle B_\alpha(s) B_\beta(0) \rangle = \frac{1}{2}\gamma_{\alpha\beta}(\omega) + iS_{\alpha\beta}(\omega), \tag{75}$$

where $B_\alpha(t) = e^{iH_B t} B_\alpha e^{-iH_B t}$, and $\gamma_{\alpha\beta}(\epsilon)$ and $S_{\alpha\beta}(\epsilon)$ are Hermitian matrices. We assume that the bath is in thermal equilibrium $\rho_B \propto e^{-\beta H_B}$ at inverse temperature $\beta$, which implies the Kubo-Martin-Schwinger (KMS) condition

$$\gamma_{\alpha\beta}(-\omega) = e^{-\beta\omega}\gamma_{\beta\alpha}(\omega). \tag{76}$$

The system-bath coupling acts on the system through $A_\alpha$ whose natural basis is the Floquet eigenstates:

$$i\frac{d}{dt}\left|\psi_m(t)\right\rangle = H_S(t)\left|\psi_m(t)\right\rangle; \quad \left|\psi_m(t)\right\rangle = e^{-i\epsilon_m t}\left|u_m(t)\right\rangle; \quad \left|u_m(t+T)\right\rangle = \left|u_m(t)\right\rangle, \tag{77}$$

where $\epsilon_m$ is called the quasienergy.

Under appropriate Born and Markov approximations [41, 85] (see also Appendix D), we obtain the Floquet-Lindblad equation with the Lindbladian

$$\mathcal{L}_t(\rho) = -i[H_S(t) + \Lambda^{\text{LS}}(t), \rho] + \mathcal{D}_t(\rho) \tag{78}$$

with the Lamb shift $\Lambda^{\text{LS}}(t)$ and the dissipator $\mathcal{D}_t$ given by

$$\Lambda^{\text{LS}}(t) = \sum_{\alpha,\beta,\epsilon} S_{\alpha\beta}(\epsilon) A_\epsilon^{\alpha\dagger}(t) A_\epsilon^\beta(t), \tag{79}$$

$$\mathcal{D}_t(\rho) = \sum_{\alpha,\beta,\epsilon} \gamma_{\alpha\beta}(\epsilon)\left[A_\epsilon^\beta(t)\rho A_\epsilon^{\alpha\dagger}(t) - \frac{1}{2}\left\{A_\epsilon^{\alpha\dagger}(t)A_\epsilon^\beta(t), \rho\right\}\right]. \tag{80}$$

Here, the jump operators are given by

$$A_\epsilon^\alpha(t) = e^{-i\epsilon t}\sum_{m,n} \mathsf{A}_{mn}^\alpha(\epsilon)\left|\psi_m(t)\right\rangle\left\langle\psi_n(t)\right|, \tag{81}$$

where the matrix elements $\mathsf{A}_{mn}^\alpha(\epsilon)$ are defined by the Fourier expansion

$$\langle\psi_m(t)|A_\alpha|\psi_n(t)\rangle = \sum_\epsilon \mathsf{A}_{mn}^\alpha(\epsilon)e^{-i\epsilon t}. \tag{82}$$

and periodic in time as $A_\epsilon^\alpha(t)$ are periodic. Since $\langle\psi_m(t)|A_\alpha|\psi_n(t)\rangle = e^{i(\epsilon_m - \epsilon_n)t}\langle u_m(t)|A_\alpha|u_n(t)\rangle$ and $|u_n(t+T)\rangle = |u(t)\rangle$, the sums over $\epsilon$ in the above equations are taken for

$$\epsilon = \epsilon_{nm;k} \equiv \epsilon_n - \epsilon_m + k\omega \quad (k \in \mathbb{Z}). \tag{83}$$

Thus, one can also rewrite Eq. (82) as

$$\langle \psi_m(t)|A_\alpha|\psi_n(t)\rangle = \sum_k \mathsf{A}^\alpha_{mn}(\epsilon_n - \epsilon_m + k\omega)\mathrm{e}^{-\mathrm{i}(\epsilon_n - \epsilon_m + k\omega)t} \equiv \sum_k \mathsf{A}^\alpha_{mn;k}\mathrm{e}^{-\mathrm{i}(\epsilon_n - \epsilon_m + k\omega)t}. \quad (84)$$

As shown in Sec. D, the jump operator $A^\alpha_\epsilon(t)$ $(A^{\alpha\dagger}_\epsilon(t))$ lowers (raises) quasienergy by $\epsilon$.

One remarkable property of this FLE is the existence of a reference frame in which the Lindbladian is time-independent. This frame is the interaction picture as is evident in the derivation of the FLE (see Appendix D). Also, as shown in Appendix D, the Lamb shift can be written, without loss of generality, as $\Lambda^{\mathrm{LS}}(t) = \sum_n \lambda_n |\psi_n(t)\rangle \langle\psi_n(t)|$, where $\lambda_n$ are the (real) eigenvalues of the Hermitian matrix $\sum_{\alpha,\beta,\epsilon,l} S_{\alpha\beta}(\epsilon)\mathsf{A}^{\alpha*}_{ml}(\epsilon)\mathsf{A}^\beta_{ln}(\epsilon)$. Using this expression for $\Lambda^{\mathrm{LS}}(t)$ and writing $\rho(t)$ as

$$\rho(t) = \sum_{m,n} |\psi_m(t)\rangle \, \sigma_{mn}(t) \, \langle\psi_n(t)|, \quad (85)$$

we have the Lindblad equation for the matrix $\sigma(t)$ (in the interaction picture):

$$\frac{d\sigma(t)}{dt} = -i[\lambda, \sigma] + \sum_{\alpha,\beta,\epsilon} \gamma_{\alpha\beta}(\epsilon)\left[\mathsf{A}^\beta(\epsilon)\sigma(t)\mathsf{A}^{\alpha\dagger}(\epsilon) - \frac{1}{2}\left\{\mathsf{A}^{\alpha\dagger}(\epsilon)\mathsf{A}^\beta(\epsilon), \sigma(t)\right\}\right], \quad (86)$$

where $(\lambda)_{mn} \equiv \delta_{mn}\lambda_m$. Thus, written for $\sigma(t)$, the Lindbladian is time-independent. This is a special property of the FLE of the RWA and not shared with other FLEs in general.

In the rest of this subsection 6.1.1, we assume that the quasienergies are not degenerate and analyze the NESS using the FLE in the time-independent frame (86). Then, the diagonal elements, $P_n(t) \equiv \sigma_{nn}(t)$, form a closed set of differential equations (see Appendix D.3 for the derivation)

$$\frac{dP_n(t)}{dt} = \sum_m [W_{nm}P_m(t) - W_{mn}P_n(t)] \quad (87)$$

with

$$W_{nm} \equiv \sum_{\alpha,\beta,\epsilon} \gamma_{\alpha\beta}(\epsilon)[\mathsf{A}^\beta(\epsilon)]_{nm}[\mathsf{A}^{\alpha\dagger}(\epsilon)]_{mn} = \sum_{\alpha,\beta,\epsilon} \gamma_{\alpha\beta}(\epsilon)\mathsf{A}^{\alpha*}_{nm}(\epsilon)\mathsf{A}^\beta_{nm}(\epsilon) \quad (88)$$

$$= \sum_{\alpha,\beta,k} \gamma_{\alpha\beta}(\epsilon_m - \epsilon_n - k\omega)\mathsf{A}^{\alpha*}_{nm;k}\mathsf{A}^\beta_{nm;k} \quad (89)$$

where we have used Eq. (84) to obtain the last line. As discussed below, the off-diagonal elements, forming another closed set of equations, do not contribute to the NESS in most cases, and we ignore them at this moment (see Appendix D.3 for detail). Since $\gamma(\epsilon)$ is a positive Hermitian matrix, $W_{nm}$ is nonnegative for any $m$ and $n$ and hence can be regarded as the transition rates from the Floquet state $|\psi_m(t)\rangle$ to another $|\psi_n(t)\rangle$. Equation (87) dictates that the Floquet-state population $P_n(t)$ obeys the classical master equation with the transition rates $W_{mn}$.

Here we make a physical interpretation of the transition rate $W_{nm}$. Equation (89) dictates that the transition rate $W_{nm}$ consists of contributions labeled by integer $k$. For each $k$, in addition to the transition amplitude $\mathsf{A}^\beta_{nm;k}$, there appears the bath's spectral weight $\gamma_{\alpha\beta}(\epsilon_m -$

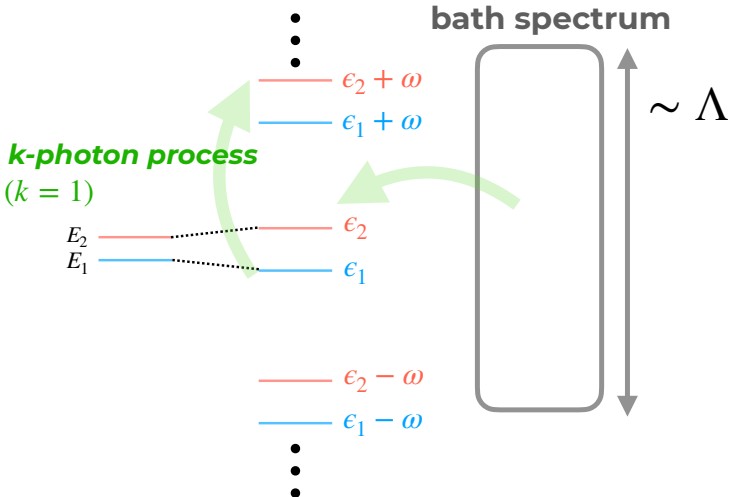

Figure 6: Schematic illustration of the $k$-photon process for $k = 1$ in a periodically driven two-level system. The curved arrow indicates a dissipation-induced transition between the Floquet states $m = 1$ and $n = 2$ entailing the quasienergy change $\epsilon_2 + \omega - \epsilon_1$. Upon this process, the bath's energy changes by $\epsilon_1 - \epsilon_2 - \omega$, which compensates the system's quasienergy change.

$\epsilon_n - k\omega$). For $k = 0$, the argument of the spectral weight is $\epsilon_m - \epsilon_n$ corresponding to the quasienergy difference between the initial $|\psi_m(t)\rangle$ and final $|\psi_n(t)\rangle$ Floquet states in the transition. For $k \neq 0$, the argument involves an extra energy $k\omega$, which corresponds to the energy of $k$ photons. From these observations, we interpret that $k$ in Eq. (89) stands for the number of photons involved in the Floquet-state transitions, and the quasienergy difference and the photon energy are exchanged between the system and bath (see Fig. 6 for illustration). Thus, in the following, we will refer to each $k$ contribution in the sum of Eq. (89) as the $k$-photon process.

To study the NESS, we assume that $W_{mn}$ is irreducible so that the classical master equation (87) has the unique steady-state solution $P_n^{\text{ss}}$, which $P_n(t)$ approaches as $t \to \infty$. This solution is characterized by Eq. (87) as

$$\sum_m (W_{nm} P_m^{\text{ss}} - W_{mn} P_n^{\text{ss}}) = 0, \tag{90}$$

which is equivalent to

$$\sum_m S_{nm} P_m^{\text{ss}} = 0, \qquad S_{nm} \equiv W_{nm} - \delta_{mn} \sum_m W_{mn}. \tag{91}$$

Equation (91) means that $P_m^{\text{ss}}$ is the zero-eigenvalue eigenvector for the matrix $S_{mn}$. Let us also assume that the NESS density matrix is unique[4], in which case all the off-diagonal

---

[4]This assumption corresponds to the one that ensures the uniqueness of the NESS eigenvector $\eta$ mentioned in Sec. 4.

elements $\sigma_{m \neq n}$ are guaranteed to vanish in the NESS (see Appendix D.3 for the proof). Once having $P_n^{\mathrm{ss}}$, we obtain the NESS density matrix as

$$\rho_{\mathrm{ness}}(t) = \sum_n P_n^{\mathrm{ss}} |\psi_n(t)\rangle \langle \psi_n(t)| . \tag{92}$$

It is noteworthy that $P_n^{\mathrm{ss}}$ and hence $\rho_{\mathrm{ness}}(t)$ are independent of the system-bath-coupling strength. Suppose that we multiply a common real number $c$ onto $A_\alpha$ (or $B_\alpha$). This results in the change $W_{nm} \to c^2 W_{nm}$. However, this change does not affect Eq. (91), meaning that $P_n^{\mathrm{ss}}$ is independent of $c$. This independence is a special property of the FLE obtained by the RWA and does not hold in general FLEs such as time-independent ones discussed in Sec. 5.

### 6.1.2 Conditions for Floquet-Gibbs state (FGS)

Before analyzing concrete examples, we generally study whether the NESS coincides with the so-called Floquet-Gibbs state (FGS)

$$\rho_{\mathrm{FG}}(t) = \sum_n P_n^{\mathrm{FG}} |\psi_n(t)\rangle \langle \psi_n(t)| ; \qquad P_n^{\mathrm{FG}} \equiv \frac{1}{Z} \sum_n e^{-\beta \epsilon_n} ; \qquad Z \equiv \sum_n e^{-\beta \epsilon_n} \tag{93}$$

for high-frequency drives. The difference between the exact NESS (92) and FGS (93) is only the Floquet-state population: While $P_n^{\mathrm{ss}}$ are determined microscopically by $W_{mn}$ as in Eq. (91), $P_n^{\mathrm{FG}}$ are the simple canonical distribution for the quasienergies. As discussed above, the quasienergies $\epsilon_m$ are defined modulo $\omega$, and the FGS is ill-defined in general. However, when $\omega$ is much larger than the system's energy scale, we have a natural choice in that $-\omega/2 < \epsilon_n < \omega/2$ and $\epsilon_n$ approaches, in the limit of $\omega \to \infty$, each eigenenergy of the time-averaged Hamiltonian. When we discuss the FGS in this paper, we implicitly assume those situations.

The FGS well approximates the NESS if the driving frequency $\omega$ is not only larger than the system's energy scale but also than the high-frequency cutoff $\Lambda$ of the bath spectral function. For simplicity, we here discuss the case in that the quasienergies $\epsilon_m$ are not degenerate and make use of the classical master equation (87). However, we can generalize the arguments to the case of degenerate quasienergies (see Appendix D.4 for detail). When $\omega$ is much larger than the bath spectral cutoff $\Lambda$, for, e.g., the ohmic bath (67), we have $\gamma(\epsilon_m - \epsilon_n + k\omega) \propto \exp\left[-\frac{(\epsilon_m - \epsilon_n + k\omega)^2}{2\Lambda^2}\right] \approx \exp\left(-\frac{k^2 \omega^2}{2\Lambda^2}\right) \approx 0$ (similar arguments hold for any other spectral functions). Thus, Eq. (89) can be approximated as

$$W_{nm} \approx \sum_{\alpha,\beta} \gamma_{\alpha\beta}(\epsilon_m - \epsilon_n) \mathsf{A}_{nm;0}^{\alpha*} \mathsf{A}_{nm;0}^{\beta} . \tag{94}$$

Within this approximation, we obtain the detailed balance condition $W_{nm} = e^{\beta(\epsilon_m - \epsilon_n)} W_{mn}$ with the help of the KMS condition (76). The detailed balance condition means that the FGS population $P_n^{\mathrm{FG}} = e^{-\beta \epsilon_n}/Z$ satisfies Eq. (91), meaning that the FGS (93) is the NESS $\rho_{\mathrm{ness}}(t) \approx \rho_{\mathrm{FG}}(t)$.

Physically speaking, all $k$-photon processes except $k = 0$ are negligible in this limiting case of $\omega \gg \Lambda$. As discussed in Sec. 6.1.1, for a $k$-photon process to occur, the accompanied photon energy $k\omega$ has to be compensated by the bath. However, if the bath spectral cutoff $\Lambda$ is much smaller than the photon energy $\omega$, this compensation is impossible.

However, in more realistic situations with $\omega \lesssim \Lambda$, $k$-photon ($k \neq 0$) processes can be relevant. As one can check easily, if we do not ignore $k \neq 0$ terms in $W_{mn}$ (89), the detailed balance condition is not satisfied, and the FGS is not necessarily the NESS except for some special cases [89].

### 6.1.3 Implementation of high-frequency expansion

Let us now discuss how we use the HF expansion approach to obtain the NESS (92). As the standard HF expansion applies to $|\psi_n(t)\rangle$, we consider how to calculate $P_n^{\text{ss}}$. Since $P_n^{\text{ss}}$ are given as the zero-eigenvalue eigenvector for $S_{mn}$ as in Eq. (91), it is sufficient to have $W_{mn}$ by the HF expansion.

First, we consider the case of $\omega \gg \Lambda$. In this case, as we have discussed in Sec. 6.1.2, we can ignore $k \neq 0$ terms in Eq. (89), having Eq. (94). Once we insert the standard $\omega^{-1}$ expansion for the quasienergies $\epsilon_m$ and $A_{nm}^{\alpha}$, we obtain the HF expansion for $W_{nm}$ and hence $P_n^{\text{ss}}$. As shown in Sec. 6.1.2, the first-order approximation in this method leads to the FGS as the NESS, and higher-order calculations give more accurate NESSs.

Second, we consider the other case $\omega \lesssim \Lambda$. In contrast to the first case, in Eq. (89), $k$'s with $|k|\omega \lesssim \Lambda$ give nonnegligible contributions to $W_{mn}$ that cannot be expanded for $\omega^{-1}$ in general. Thus, we have

$$W_{nm} \approx \sum_{\alpha,\beta} \sum_{k=-k_c}^{k_c} \gamma_{\alpha\beta}(\epsilon_m - \epsilon_n + k\omega) A_{nm}^{\alpha*}(\epsilon_m - \epsilon_n + k\omega) A_{nm}^{\beta}(\epsilon_m - \epsilon_n + k\omega), \qquad (95)$$

where $k_c > 0$ is a cutoff for the photon number $k$ and depends on the accuracy which we require for the results. Within the approximation (95), we can use the HF expansion for the quasienergies $\epsilon_m$ and $A_{nm}^{\alpha}$, obtaining a good approximation for $W_{nm}$ and hence $P_n^{\text{ss}}$.

In the following sections, we demonstrate that this method quantitatively works in concrete models and show that the FGS may not be the NESS in some cases.

### 6.2 Example 3: Three-level system revisited and deviation from FGS

Now we study concrete example models and their NESSs for time-dependent dissipators. As a first example, we revisit the three-level system studied in Sec. 5.3, making the dissipator time-dependent within the RWA. For simplicity, we neglect the Lamb shift. In the absence of nematic terms $N_z$ and $N_{xy}$, this model has recently been analyzed for a general spin $S$, and the environment-controlled Floquet-state paramagnetism has been proposed [90].

Following Eq. (80), we write down the time-dependent dissipator. For concrete calculations, we use $|u_m(t)\rangle$ instead of $|\psi_m(t)\rangle$. In doing so, we choose $\epsilon_m$ so that $\epsilon_m$ satisfy $E_m - \Omega/2 \leq \epsilon_m < E_m + \Omega/2$ for $m = 1, 2$, and 3. Note that this choice is possible only for high-enough frequencies that we consider here. We also define the set of possible values for quasienergy differences as

$$S_{\delta\epsilon} = \{\mathcal{E} = \epsilon_n - \epsilon_m \mid 1 \leq m, n \leq 3\}. \qquad (96)$$

Note that this set has 7 elements since $\epsilon_m$ are not degenerate in our model. Then the dissipator is given by

$$\mathcal{D}_t(\rho) = \sum_{\mathcal{E} \in S_{\delta\epsilon}} \sum_{k=-\infty}^{\infty} \gamma(\mathcal{E} + k\omega) \left[ A_{\mathcal{E},k}(t) \rho A_{\mathcal{E},k}^{\dagger}(t) - \frac{1}{2} \left\{ A_{\mathcal{E},k}^{\dagger}(t) A_{\mathcal{E},k}(t), \rho \right\} \right], \qquad (97)$$

where

$$A_{\mathcal{E},k}(t) = e^{ik\omega t} \sum_{\substack{m,n \\ (\epsilon_n - \epsilon_m = \mathcal{E})}} \mathsf{A}_{mn;k} \,|u_m(t)\rangle \langle u_n(t)| \,, \tag{98}$$

$$\mathsf{A}_{mn;k} = \int_0^T \frac{\mathrm{d}t}{T} e^{ik\omega t} \langle u_m(t)|S_x|u_n(t)\rangle \,. \tag{99}$$

As remarked in Sec. 6.1, we can utilize, for this class of problems, a convenient frame in which the Lindbladian is time-independent. Thus, we make use of it and consider Eq. (88) that leads to

$$W_{mn} = \sum_{k=-\infty}^{\infty} \gamma(\epsilon_n - \epsilon_m + k\omega)|\mathsf{A}_{mn;k}|^2, \tag{100}$$

which gives the Floquet-state population $P_n^{\mathrm{ss}}$ by Eq. (92).

Our aim here is to show how well the HF-expansion approach formulated in Sec. 6.1.3 gives a systematic approximation for $\rho_{\mathrm{ness}}(t)$. Let us suppose that we try to obtain $P_n^{\mathrm{ss}}$ up to $O(\omega^{-N})$ for an $N \,(\geq 0)$. For this, we invoke the HF expansion for isolated systems to have the Floquet states $|u_n(t)\rangle$ and their quasienergies $\epsilon_m$ up to this order, which give $\mathsf{A}_{mn;k}$ up to the desired order $O(\omega^{-N})$. More concretely, we calculate $H_{\mathrm{eff}}$ and $K(t)$ up to $O(\omega^{-N})$ by the van Vleck expansion (12), which give $|u_n(t)\rangle = e^{-iK(t)}|E_n^{\mathrm{eff}}\rangle$ and $\epsilon_n = E_n^{\mathrm{eff}}$, where $H_{\mathrm{eff}}|E_n^{\mathrm{eff}}\rangle = E_n^{\mathrm{eff}}|E_n^{\mathrm{eff}}\rangle$. Although these calculations provide appropriate approximations for each term in the summation of Eq. (100), we need to take a large-enough cutoff $k_c$ for

$$W_{mn} \approx \sum_{k=-k_c}^{k_c} \gamma(\epsilon_n - \epsilon_m + k\omega)|\mathsf{A}_{mn;k}|^2 \tag{101}$$

so that the truncation error can be negligible in our $O(\omega^{-N})$ calculation. For the present model, the appropriate choice is $k_c = \lfloor (N+1)/2 \rfloor$ ($\lfloor \cdots \rfloor$ denotes the floor function) because of the following reason. As one can check easily in our model, $|\mathsf{A}_{mn;k}|^2 = O(\omega^{-2|k|})$, whereas $\gamma(\epsilon_n - \epsilon_m + k\omega)$ is at most $O(\omega^1)$ for $k \neq 0$, and their product is of $O(\omega^{-2|k|+1})$. Thus, once we choose $k_c = \lfloor (N+1)/2 \rfloor$, the truncation error (i.e., the difference between Eqs. (100) and (101)) is negligible in our $O(\omega^{-N})$ calculation. Once we have $W_{mn}$, we obtain the steady-state population $P_n^{\mathrm{ss}}$ from Eq. (91) and $\rho_{\mathrm{ness}}(t)$ by Eq. (92).

The one-cycle evolutions for the NESS calculated in this way and in the exact numerical time integration are plotted in Fig. 7. As in Sec. 5.3, we use the bath parameters as $\beta = 3$, and $\Lambda = 10$ (the NESS does not depend on $\gamma_0$ in the present case). For the HF approach, we plot the results for the zeroth, first, and second orders, which are obtained by expanding $H_{\mathrm{eff}}$ and $K(t)$ up to $O(\omega^{-N})$ for $N = 0, 1$, and 2, respectively. It is worth noting the similarity between Figs. 7 and 3. This means that the approximate bath model in Sec. 5.3 is quantitatively good for high-frequency drives.

Similarly to the time-independent dissipator studied in Sec. 5.3, at the zeroth order, the NESS corresponds to the thermal equilibrium showing no time dependence. As we increase the order, the HF result tends to approach the exact numerical integration, which we verify quantitatively below. We note that there are some period-$T/2$ (or the second harmonic) oscillations in, e.g., $S_x^2 - S_y^2$ in Fig. 3. These oscillations cannot be taken up to the first order

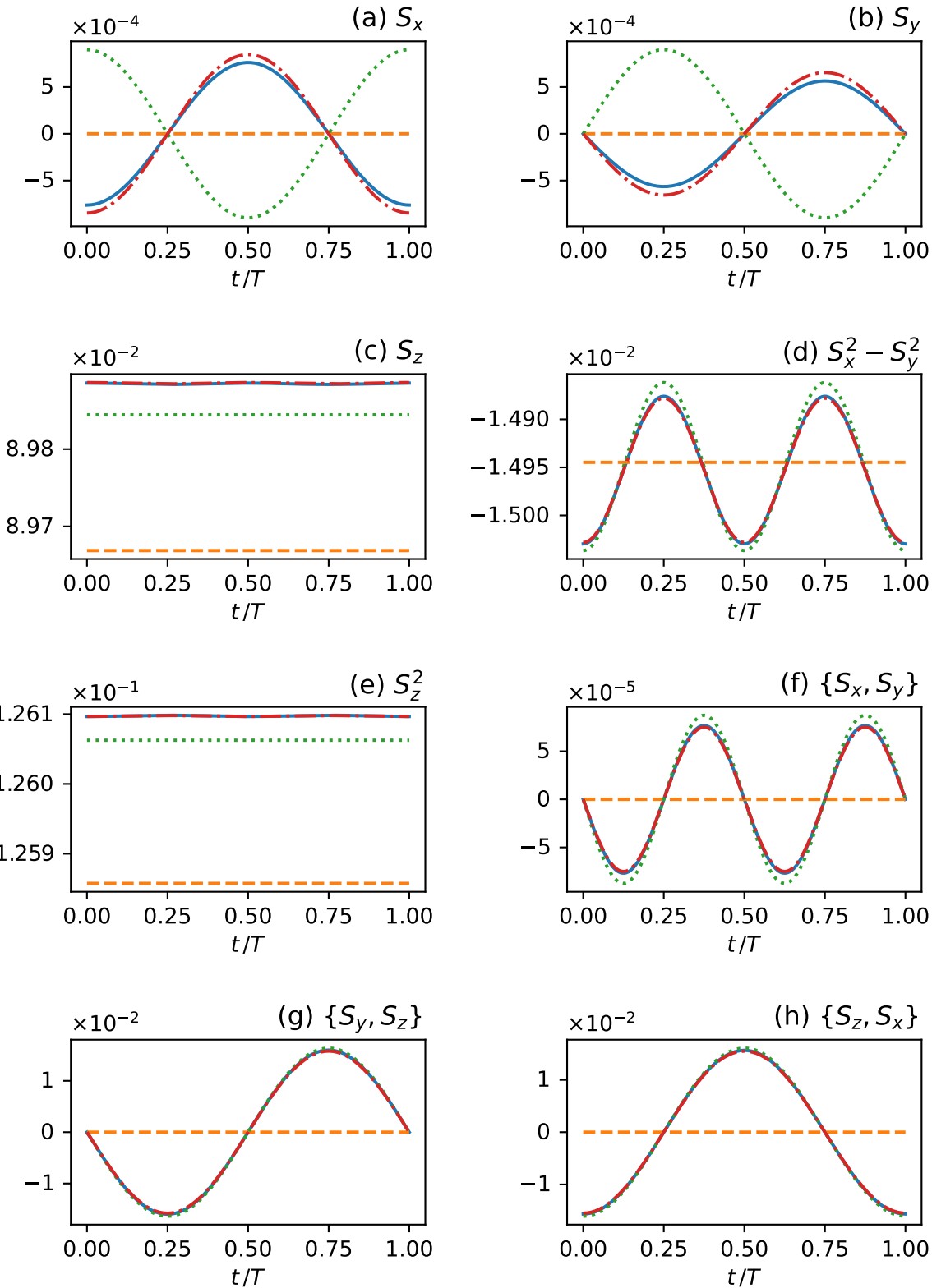

Figure 7: NESS in one cycle calculated by the exact numerical integration (blue) and by the HF expansion approach at order $N = 0$ (orange), 1 (green), and 2 (red). All the independent 8 observables are plotted in each panel. For each $N$, we expand $H_{\text{eff}}$ and $K(t)$ up to $O(\omega^{-N})$ and set $k_c = \lfloor (N+1)/2 \rfloor$. The parameters are chosen as $B_s = 0.3$, $N_z = 1$, $N_{xy} = 0.05$, $B_d = 0.1$, $\omega = 10$, $\beta = 3$, $\gamma_0 = 0.2$, and $\Lambda = 10$.

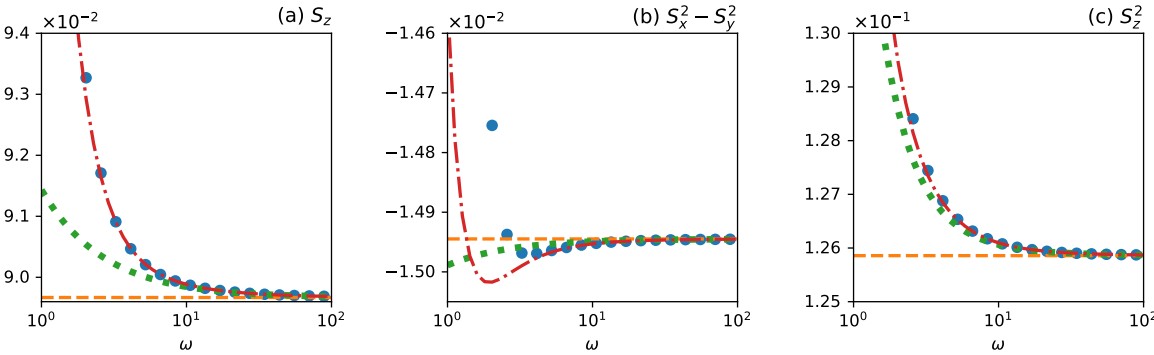

Figure 8: One-cycle averages of observables in the NESS [Eq. (92)] plotted against $\omega$. Here $\rho_{\mathrm{ness}}(t)$ is calculated by the exact numerical integration (blue) and by the HF expansion approach at order $N = 0$ (orange), 1 (green), and 2 (red). For each $N$, we expand $H_{\mathrm{eff}}$ and $K(t)$ up to $O(\omega^{-N})$ and set $k_c = \lfloor (N + 1)/2 \rfloor$. The parameters are the same as in Fig. 7 except $\omega$.

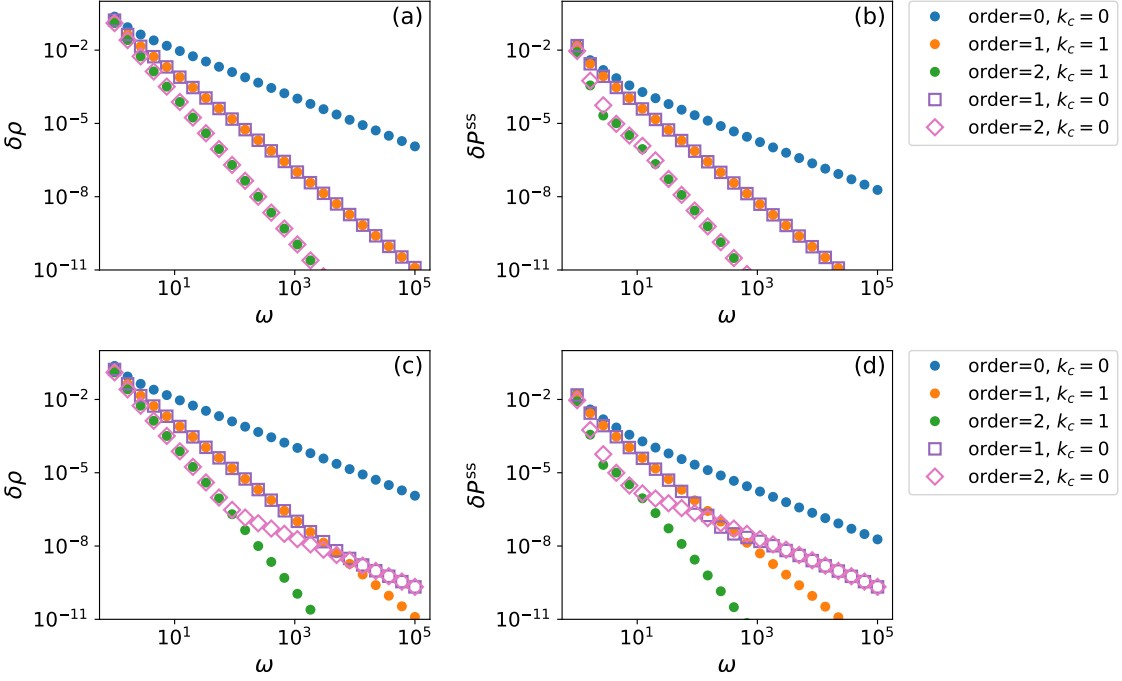

Figure 9: Accuracy of HF expansion approach for the NESS density matrix $\delta\rho_N$ (73) (left panels) and the Floquet-state population $\delta P_N^{\mathrm{ss}}$ (105) (right panels) plotted against $\omega$ for the bath spectral cutoffs $\Lambda = 10$ (upper panels) and $10^6$ (lower panels). Each data set corresponds to a pair of $(N, k_c)$ as shown in legend, for which we use $H_{\mathrm{eff}}$ and $K(t)$ expanded up to $O(\omega^{-N})$. The other parameters are chosen as $B_s = 0.3$, $N_z = 1$, $N_{xy} = 0.05$, $B_d = 0.1$, $\omega = 10$, and $\beta = 3$.

since the micromotion $K^{(1)}(t)$ contains only frequency $\omega$. At the second order, $\rho_{\text{ness}}^{(N=2)}(t) = e^{\mathcal{G}_t}\eta = (1 + \mathcal{G}_t + \frac{1}{2}\mathcal{G}_t^2)\eta$, where $\frac{1}{2}\mathcal{G}_t^2\eta$ involves the frequency $2\omega$.

The HF expansion approach becomes more accurate for higher frequency $\omega$. Figure 8 shows the $\omega$-dependence of one-cycle averages (72) of observables $O = S_z, S_x^2 - S_y^2$, and $S_z^2$ in the NESS as the other observables give vanishing averages due to symmetry reasons. For a more strict measure of the accuracy of the HF approach, we again consider $\delta\rho_N$ defined in Eq. (73). Figure 9(a) shows $\delta\rho$ for the orders $0, 1$, and $2$ plotted against $\omega$. We here observe $\delta\rho_N \propto \omega^{-(N+1)}$ for high frequencies. This is a clear indication that the $N$-th order approximation $\rho_{\text{ness}}^{(N)}(t)$ completely describe $\rho_{\text{ness}}^{\text{exact}}(t)$ up to $O(\omega^{-N})$.

We remark the qualitative difference between the time-independent and time-dependent dissipators found in the one-cycle average in a quadrupolar (spin-nematic) operator $\{S_x, S_y\} = \frac{1}{2i}((S^+)^2 - (S^-)^2)$ [91–94]: While it does not vanish for the time-dependent dissipator, it does for the time-independent one. As shown in Ref. [57], one can understand this difference by the following antiunitary symmetry operator $V$: $VS_yV^\dagger = -S_y$ and $VS_\alpha V^\dagger = S_\alpha$ ($\alpha = x$ and $z$). One can easily check that our Hamiltonian satisfies the following dynamical symmetry $H_{\text{NV}}(t) = VH_{\text{NV}}(T-t)V^\dagger$. This means that $|\tilde{u}_n(t)\rangle \equiv V|u_n(T-t)\rangle$ is also a Floquet state having the same quasienergy with $|u_n(t)\rangle$. If there is no degeneracy in quasienergies as is the case of our model, $|\tilde{u}_n(t)\rangle$ and $|u_n(t)\rangle$ are the same state: there exists a phase factor $e^{i\theta_n}$ ($\theta_n \in \mathbb{R}$) such that $|\tilde{u}_n(t)\rangle = e^{i\theta_n}|u_n(t)\rangle$. Noticing $V^\dagger\{S_x, S_y\}V = -\{S_x, S_y\}$, we have

$$\overline{\langle u_n(t)|\{S_x, S_y\}|u_n(t)\rangle} = \overline{\langle \tilde{u}_n(t)|\{S_x, S_y\}|\tilde{u}_n(t)\rangle} \tag{102}$$

$$= -\overline{\langle u_n(T-t)|\{S_x, S_y\}|u_n(T-t)\rangle} \tag{103}$$

$$= -\overline{\langle u_n(t)|\{S_x, S_y\}|u_n(t)\rangle}, \tag{104}$$

which means $\overline{\langle u_n(t)|\{S_x, S_y\}|u_n(t)\rangle} = 0$. Therefore, from the definition of $\rho_{\text{ness}}(t)$ in Eq. (92), we have $\overline{\{S_x, S_y\}} = \overline{\text{tr}[\rho_{\text{ness}}(t)\{S_x, S_y\}]} = \sum_n P_n^{\text{ss}}\overline{\langle u_n(t)|\{S_x, S_y\}|u_n(t)\rangle} = 0$. In words, the Hamiltonian's antiunitary symmetry is not broken by the dissipation if it is treated within the RWA. This is a special property of this class of FLEs that become time-independent in the interaction picture. In general, antiunitary symmetries can be broken as in the time-independent dissipator that we analyze in Sec. 5.3.

Let us discuss the cutoff $\Lambda$ of the bath spectral function. We have thus far set $\Lambda = 10$ and confirmed that the HF approach works well for $\omega \gtrsim \Lambda = 10$. For $\omega \gtrsim \Lambda$, the subtlety about the cutoff $k_c$ discussed in Sec. 6.1.3 has not been problematic, and $k_c$ has not played important roles. In fact, in Fig. 9(a), we also plot $\delta\rho_N$ for $N = 1$ and $2$ with $k_c = 0$ that turn out to coincide with those with $k_c = 1$. As another measure of the HF approach's accuracy, we introduce

$$\delta P_N^{\text{ss}} \equiv \sqrt{\sum_{n=1}^{3}(P_{N,n}^{\text{ss}} - P_n^{\text{ss}})^2}, \tag{105}$$

where $P_{N,n}^{\text{ss}}$ denotes $P_n^{\text{ss}}$ obtained by our HF approach at the $N$-th order whereas $P_n^{\text{ss}}$ does the numerical exact solution. In Fig. 9(b), we plot $\delta P_N^{\text{ss}}$ for $N = 0, 1$ and $2$ with $k_c = 0$ and $1$ and find that the choice of $k_c$ is not relevant. We do not have to set $k_c$ to be larger than zero because each contribution from $k \neq 0$ in Eq. (100) is negligibly small as $\gamma(\epsilon_n - \epsilon_m + k\omega) \approx 0$ if $\omega \gg \Lambda$.

A more nontrivial situation happens when $\omega \ll \Lambda$. To verify how our HF approach works in this case, we perform a similar numerical analysis for $\Lambda = 10^6$ with all the other parameters

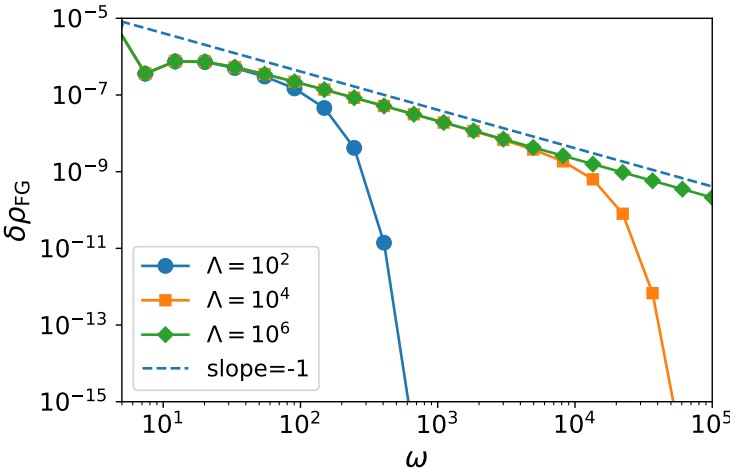

Figure 10: Difference between the FGS (93) and the exact NESS (92) for bath spectral cutoff $\Lambda = 10^2$ (circle), $10^4$ (square), and $10^6$ (diamond). The dashed line is the guide to the eye showing the slope of $-1$. The other parameters are chosen as $B_s = 0.3$, $N_z = 1$, $N_{xy} = 0.05$, $B_d = 0.1$, $\omega = 10$, and $\beta = 3$.

being the same. Figures 9(c) and (d) show $\delta \rho_N$ and $\delta P_N^{\mathrm{ss}}$ for orders $N = 0, 1$, and 2 with cutoff $k_c = 0$ and 1. For the correct cutoffs $k_c = 0$ for $N = 0$ and $k_c = 1$ for $N = 1$ and 2, we obtain $\delta \rho_N \propto \delta P_N^{\mathrm{ss}} \propto \omega^{-N-1}$, meaning that the $N$-th order HF expansion correctly describes the NESS up to $O(\omega^{-N})$. However, when we set the smaller cutoff $k_c = 0$ for $N = 1$ and 2, $\delta \rho_N$ and $\delta P_N^{\mathrm{ss}}$ eventually scales like $\omega^{-1}$ as $\omega$ increases. For this range of $\omega$, by taking $k_c$ appropriately, we can obtain the NESS at a desired precision with the HF expansion.

Finally, we comment on the accuracy of the FGS, which we quantify by

$$\delta \rho_{\mathrm{FG}} \equiv \| \rho_{\mathrm{FG}}(t) - \rho_{\mathrm{ness}}(t) \|. \tag{106}$$

Equations (92) and (93) lead to $\rho_{\mathrm{FG}} = \sqrt{\sum_n (P_n^{\mathrm{FG}} - P_n^{\mathrm{ss}})^2}$, which is time-independent. We plot $\delta \rho_{\mathrm{FG}}$ in Fig. 10 for different bath spectral cutoffs $\Lambda$. We note that, to calculate the FGS exactly, we need numerical integration to obtain the quasienergies $\epsilon_n$ and time-dependent Floquet states $|\psi_n(t)\rangle$. As discussed in Sec. 6.1.2, the FGS becomes accurate rapidly when $\omega$ exceeds $\Lambda$, where $k$-photon ($k \neq 0$) processes are suppressed. Meanwhile, for $\omega \lesssim \Lambda$, the FGS's error is as large as $O(\omega^{-1})$, which derives from the 1-photon processes of $O(\omega^{-1})$ neglected in the FGS. Thus, for $\omega \lesssim \Lambda$, the HF expansion approach with appropriately taking $k$-photon process gives better description for the NESS without numerical integration. In the following section, we further investigate the NESS and FGS in another model.

## 6.3 Example 4: Inverse Faraday Effect in Heisenberg chain and dissipation-assisted Floquet engineering

Magneto-optics has been long studied actively [25, 95] and various ultrafast methods of controlling magnetism with light have been discussed. Among them, the inverse Faraday effect [23–25] is a representative phenomenon and it means a magnetization change by applying

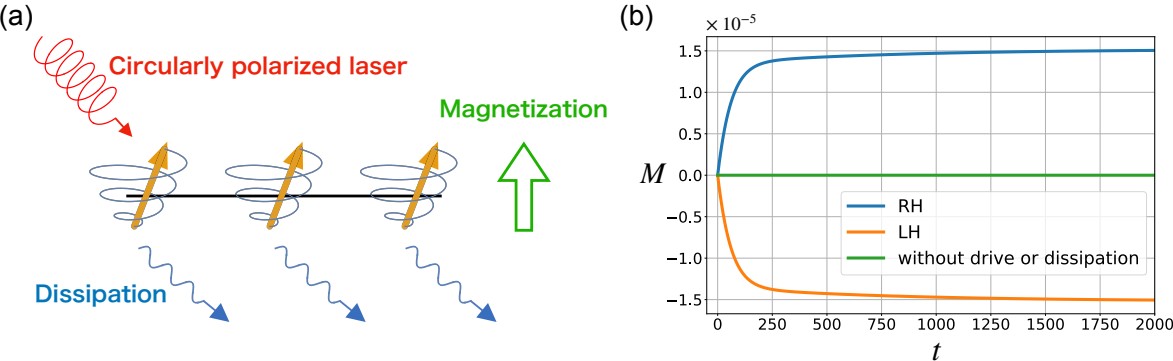

Figure 11: (a) Illustration of inverse Faraday effect in the presence of dissipation. (b) Time evolution of magnetization along $z$-axis. The blue (orange) line shows a result for a right (left)-handed circularly polarized laser. The parameters are $B = 0.1, \omega = \pm 5, \beta = 10, \Lambda = 5$, and $L = 8$.

circularly polarized light to magnetic materials. The sign of varied magnetization can be controlled by switching the polarization direction (i.e., left- or right-handed polarization). This effect can be viewed as one of typical Floquet engineerings. The inverse Faraday effect has been indeed observed in various magnetic materials [96–100] with application of visible or infrared light whose photon energy is comparable to the electron band energy.

Its microscopic theory [24] was first developed by Pershan *et al.*, who predicted circularly-polarized light-driven Zeeman coupling appears when the light is applied to conducting electron systems with spin-orbit (SO) coupling. We emphasize that an SO coupling is necessary to generate the effective Zeeman interaction between electron spins and the ac electric field of the light.

Recently, in magnetic insulators, the THz-light-driven (or shortly THz) inverse Faraday effect has been investigated theoretically [101–103]. The THz light is suitable in these systems as its photon energy is comparable to the magnetic excitation energy of, e.g., magnons and spinons, especially in antiferromagnetic systems. As in conducting electrons, magnetic anisotropy originated from the SO coupling is necessary for the THz inverse Faraday effect. Namely, the spin rotation symmetry has to be broken [102] to induce a THz-laser-driven magnetization.

On the other hand, instead of an SO coupling or a magnetic anisotropy, we expect that even dissipation can also assist the emergence of the THz inverse Faraday effect because the Lindblad type dissipation can break spin rotation symmetry. In fact, small SO coupling or spin-rotation-symmetry breaking interactions (such as spin-phonon coupling and dipole interaction) are always present even in ideal magnetic materials. Such weak but finite interactions may be viewed as a source of dissipation for the systems we consider.

To demonstrate this dissipation-assisted THz inverse Faraday effect, we consider an SU(2)-symmetric Heisenberg spin-1/2 chain driven by a circularly polarized laser. The Hamiltonian

and dissipator are given by

$$
\begin{aligned}
H(t) &= H_0 + V(t) \\
&= J \sum_{j=1}^{L} \vec{S}_j \cdot \vec{S}_{j+1} + B \sum_{j=1}^{L} \left[ S_j^x \cos(\omega t) + S_j^y \sin(\omega t) \right],
\end{aligned}
\tag{107}
$$

$$
\mathcal{D}_t(\rho) = \sum_{j,\epsilon} \gamma(\epsilon) \left[ A_\epsilon^j(t) \rho A_\epsilon^{j\dagger}(t) - \frac{1}{2} \{ A_\epsilon^{j\dagger}(t) A_\epsilon^j(t), \rho \} \right],
\tag{108}
$$

where $\vec{S}_j$ is the spin-$\frac{1}{2}$ operator on $j$th site and $J > 0$ is the antiferromagnetic exchange coupling, and $B$ is the coupling constant of the ac Zeeman interaction of the circularly polarized THz laser with frequency $\omega$. In the dissipator $\mathcal{D}_t(\rho)$, $A_\epsilon^j(t)$ is defined in Eqs. (81) and (82) with $A^j = S_j^x$. Here, as we did in other sections, we suppose that $\gamma(\epsilon)$ is the spectral function of the ohmic boson bath with a Gaussian cutoff $\Lambda$ and, for convenience, redisplay its form:

$$
\gamma(\epsilon) = \gamma_0 \frac{\epsilon e^{-\frac{\epsilon^2}{2\Lambda^2}}}{1 - e^{-\beta\epsilon}}.
\tag{109}
$$

For simplicity, we have ignored the Lamb shift $\Lambda^{\mathrm{LS}}(t)$ here, which does not change the NESS because both the Lamb shift and NESS are diagonal in the time-independent frame (see, for example, Eqs. (86) and (87)). In this subsection, we calculate the magnetization dynamics and its value in the NESS by solving the time-dependent Floquet-Lindblad equation of Eqs. (107) and (108) with the forth-order Runge-Kutta method. The initial state is the ground state of $H_0 = J \sum_{j=1}^{L} \vec{S}_j \cdot \vec{S}_{j+1}$, which is a spin singlet for even $L$'s. From now on, we fix the parameters as $J = 1$ and $\gamma_0 = 0.1$.

In the absence of the dissipation (i.e., $\gamma_0 = 0$), the inverse Faraday effect does not occur because of the spin SU(2) symmetry of $H_0$, which leads to a conservation law $[H(t), S_{\mathrm{tot}}^2] = 0$ [102]. The symmetry prohibits the magnetization amplitude from growing up, although its direction changes due to the precession around the circularly polarized laser.

On the other hand, in the presence of the dissipation (i.e., $\gamma_0 \neq 0$), the conservation law breaks down due to the dissipator, $[A_\epsilon^j(t), S_{\mathrm{tot}}^2] \neq 0$, and the total magnetization can grow up. Figure 11 shows the inverse Faraday effect by dissipation. The total magnetization $M = \sum_j \langle S_j^z \rangle / L$ starts from zero and grows up, approaching a nonzero value. We note that the right- and left-handed circularly polarized lasers give rise to the opposite directions of the magnetization because they are transformed by the $\pi$-rotation around the $x$-axis, $S_j^x \to S_j^x, S_j^y \to -S_j^y$, and $S_j^z \to -S_j^z$.

As discussed in Sec. 6.1, if the driving frequency is larger not only than the system's energy scale but also than the bath spectral cutoff $\Lambda$, the NESS in this model is approximately the Floquet-Gibbs state (93) with the effective Hamiltonian

$$
H_{\mathrm{eff}} = H_0 - \frac{B^2}{2\omega} \sum_j S_j^z + O(1/\omega^2).
\tag{110}
$$

The second term on the right-hand side in Eq. (110) is an effective magnetic field induced by the circularly polarized laser, giving rise to the inverse Faraday effect. Recall again that, in the absence of dissipation, the effective magnetic field only cannot induce magnetization, as illustrated in Fig. 11(b). Figures 12 (a)-(d) show the $\omega$-dependence of the magnetizations in

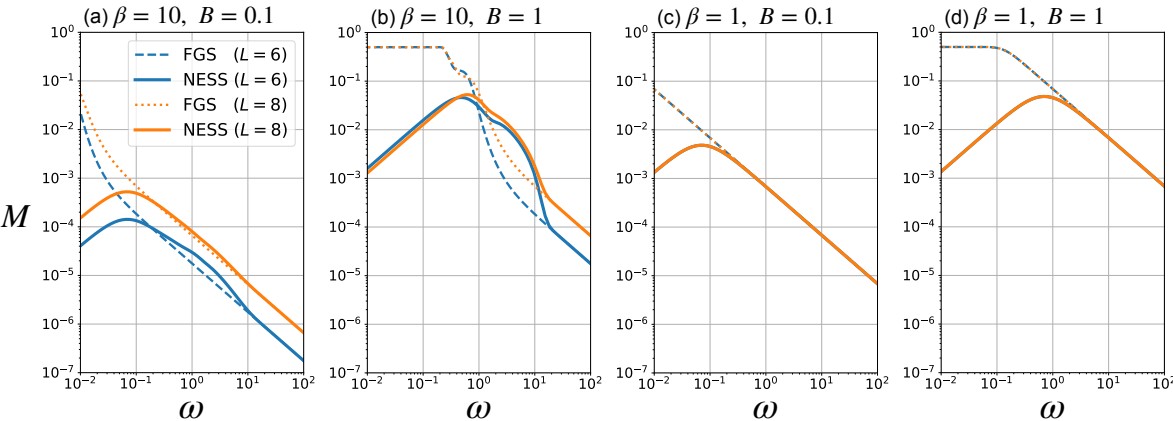

Figure 12: $\omega$-dependence of total magnetization $M = \sum_j \langle S_j^z \rangle / L$ for $(\beta, B) = $ (a)$(10, 0.1)$, (b)$(10, 1)$, (c)$(1, 0.1)$, and (d)$(1, 1)$. The dashed and dotted lines are the first-order Floquet-Gibbs state (FGS), and the blue and orange solid lines denote the exact NESS for $L = 6$ and 8. Note that the lines for $L = 6$ and 8 are overlapped in (c) and (d).

the exact NESS and FGS for various $\beta$ and $B$. For any $\beta$ and $B$, the NESS is well described by the FGS in the high-frequency region, where the magnetization decays as $M \propto \omega^{-1}$.

We briefly discuss the system size dependence although the numerically accessible system sizes are strongly limited by computational complexity in the dissipative quantum many-body system. Figure 12 shows that the high-frequency expansion predicts the NESS for each of $L = 6$ and 8. Meanwhile, we observe that the magnetization shows slow system-size convergence for lower temperatures. The reason why the system-size dependence for $\beta = 10$ is larger than that for $\beta = 1$ is that the low-temperature state is sensitive to the fictitious energy gap due to the finite-size effect (the Heisenberg chain is known to be gapless in the thermodynamic limit [104, 105]).

Another subtlety in approaching the thermodynamic limit $L \to \infty$ is the heating and the breakdown of the HF expansion. However, we do not expect that this subtlety is a serious problem in our model. This is because our Hamiltonian $H(t)$ becomes time-independent in the rotating frame $|\psi\rangle \to e^{-iS^z \omega t} |\psi\rangle$, and thus the system does not heat up. We note that, in the generic many-body systems, it is known that the high-frequency expansion does not converge in the thermodynamic limit, at least in the isolated systems [2, 31, 32]. Understanding the high-frequency expansion in the generic many-body systems with dissipation is an open question, and we do not go into detail further in this work.

Let us now study the breakdown of the FGS. As discussed in the previous section, the NESS is not necessarily well approximated by the FGS due to the spectral function $\gamma(\epsilon)$. The sufficient condition that the FGS is a good approximation is that the contributions of $k \neq 0$ in $W_{mn}$ of Eq. (89) are much smaller than that of $k = 0$ within the desired order of $\omega$. For example, when the cutoff $\Lambda$ in $\gamma(\epsilon)$ of Eq. (109) is much larger than $\omega$, we need to include the contribution of $k = 1$ to obtain the accurate first-order HF expansion because $\gamma(k\omega) \sim O(\omega)$ and $|A_{nm;k}^{\alpha*} A_{nm;k}^{\beta}| \sim O(\omega^{-2|k|})$. Then, the approximated NESS is not the FGS. Conversely, when the cutoff $\Lambda$ is much smaller than $\omega$, the contribution of $k \neq 0$ is neglectable because of $|\gamma(0)| \gg |\gamma(k\omega)|$ ($k \neq 0$) (see Fig. 13 (c) for the configuration of $\gamma(\epsilon)$ with varied $\Lambda$). Figure 13 (a) shows the $\Lambda$-dependence of the magnetization in the NESS and the FGS for various $\omega$. We observe that the FGS indeed becomes less accurate for larger $\Lambda$ and the threshold of the

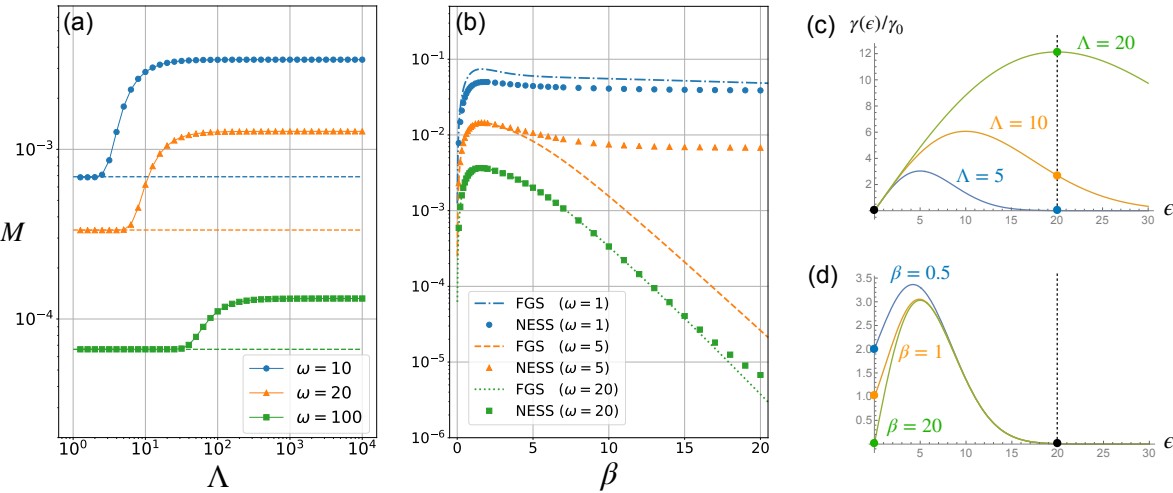

Figure 13: (a)$\Lambda$-dependence of total magnetization $M$. The circles, triangles, and squares denote the magnetization in the NESS for $\omega = 10, 20$, and $100$, respectively. The dashed lines are the magnetization in the FGS for each $\omega$. The other parameters are $B = 1, \beta = 10$, and $L = 8$. (b) $\beta$-dependence of total magnetization $M$. The circles, triangles, and squares denote the magnetization in the NESS for $\omega = 10, 20$, and $100$, and the dash-dotted, dashed, and dotted lines are the magnetization in the FGS for corresponding $\omega$. The other parameters are $B = 1, \Lambda = 5$, and $L = 8$. (c,d) $\gamma(\epsilon)/\gamma_0$ are plotted for (c) $\Lambda = 5, 10$, and $20$ with $\beta = 10$ and (d) $\beta = 0.5, 1$, and $20$ with $\Lambda = 5$. In $\gamma(\epsilon)$, the $y$-intercept is $1/\beta$ and the position of the peak is $O(\Lambda)$. Therefore, for example, $|\gamma(0)/\gamma(\omega = 20)|$ becomes smaller for the larger $\Lambda$ or the larger $\beta$.

breakdown is $O(\omega)$.

Besides, the FGS becomes less accurate for lower temperature. This is because $\gamma(k\omega)$ becomes relatively larger in comparison with $\gamma(0) = 1/\beta$ for the larger $\beta$ (see Fig. 13 (d)), meaning that we cannot neglect the contribution of $k \neq 0$ in $W_{mn}$ of Eq. (89) in the low temperature. Figure 13 (b) shows the $\beta$-dependence of the magnetization in the NESS and the FGS for various $\omega$. Indeed, the FGS becomes less accurate for larger $\beta$ and the range where the FGS is valid becomes wider as $\omega$ increases.

Finally, we emphasize that the concept of the dissipation-assisted inverse Faraday effect would be valid even in real materials as well as nanodevices consisting of a few spins although the analysis can not be quantitatively correct. The FLE with the RWA that we have used in this section is known not to work well in many-body systems in general since we cannot use the rotating wave approximation in the derivation. However, even in real materials, the dissipation can break the spin symmetry, giving rise to the inverse Faraday effect, although the detailed form of the dissipation can be different from the Lindblad-type dissipation. The detailed analysis for real materials that are not described by the Lindblad equation is future work. We also note that dissipation-assisted engineered observables can change, depending on the type of dissipation. As we discussed in Sec. 6.2 and Ref. [57], in the driven single NV-center model with the time-independent dissipator, we can find a dissipation-assisted Floquet engineering of a quadrupolar moment $\{S_x, S_y\}$, which is odd for the antiunitary symmetry operation $V$.

# 7  Conclusion

In this paper, we have theoretically studied the nonequilibrium steady states (NESSs) in the time-periodic quantum master equation of Lindblad (or GKSL) form. Considering the high-frequency regime, we have developed a systematic high-frequency (HF) expansion for Liouvillians. One important consequence is that, although the effective Liouvillian is not necessarily of Lindblad form (Lindbladian), it is still useful to analyze the NESSs. This is mainly because the effective Liouvillian is trace-preserving (Lemma 1) and the NESS is guaranteed to exist at each order of the HF expansion (Theorem 1). With the effective Liouvillian and micromotion superoperators at high-enough expansion order, we can obtain the NESS density matrix by linear algebra without directly solving the time evolution obeying the Lindblad equation.

We have first applied this theory to concrete models for one-body and many-body quantum systems subject to phenomenological time-independent dissipators. With the HF-expansion approach, the effects of external drives are manifest in the effective Liouvillian and micromotion superoperators, which are represented by the commutators of Hamiltonians and dissipators. This representation, without solving the Lindblad equation, has enabled us to grasp physical consequences of the drive and dissipation, such as the effective Hamiltonian in an NV center (Sec. 5.3) the robustness against boundary dissipation in the XY spin chain (Sec. 5.2).

We have then analyzed the microscopically-derived time-dependent dissipators within the rotation wave approximation (RWA). Besides the general theoretical framework in Secs. 3 and 4, we have developed a slightly different HF-expansion method utilizing a special property of this class of dissipators. For these dissipators realized in a weak contact to heat baths, we have shown that the cutoff $\Lambda$ of the bath spectral function serves as an important energy scale. Within these Floquet-Lindblad equations, we have generally shown that the NESS is well-described by the Floquet-Gibbs state [Eq. (93)] irrespective of model details as long as the driving frequency $(\hbar)\omega$ is higher than both the system's energy scale and the cutoff $\Lambda$. Meanwhile, when $(\hbar)\omega$ is higher than the system's energy scale but smaller than the cutoff $\Lambda$, richer physics, such as the breakdown of the Floquet-Gibbs state, happen due to dissipation processes accompanying photon exchange between the system and baths. In these cases, however, the HF-expansion approach, being implemented appropriately, enables a systematic analysis of the NESS. We have exemplified these in an NV center (Sec. 6.2) and the dissipation-assisted inverse Faraday effect in the isotropic Heisenberg spin chain.

One important open problem is whether the HF expansion makes sense in generic many-body systems, where the so-called heating problem matters. In isolated many-body systems, the periodic drive is known to heat the system to the featureless infinite-temperature state, and thus the HF-expansion description is valid at finite time range [2, 31, 32]. This phenomenon is related to the fact that the HF expansion for generic many-body systems is not a convergent series. The example systems that we have analyzed in this paper are so special that the heating does not occur. However, it is of great interest how the heating is compensated by dissipation and the convergence property of the effective Liouvillian in generic many-body systems. To address these systems, we may need to consider Floquet-Lindblad equations beyond the RWA, which is another important open problem.

# Acknowledgements

Fruitful discussions with M. Holthaus, K. Mizuta, T. Mori, F. Nathan, A. Polkovnikov, M. Rudner, T. Satoh, and H. Tsunetsugu are gratefully acknowledged.

**Funding information**   T.N.I was supported by JSPS KAKENHI Grant No. JP21K13852. K.C. was supported by JSPS KAKENHI Grant No. 21J11245 and Advanced Leading Graduate Course for Photon Science at the University of Tokyo. M.S. was supported by JSPS KAKENHI (Grants No. 17K05513 and No. 20H01830) and a Grant-in-Aid for Scientific Research on Innovative Areas "Quantum Liquid Crystals" (Grant No. JP19H05825).

# A   van Vleck high-frequency expansion in isolated systems

We recap the results of Refs. [8,9]. In an isolated system, the time evolution is given by the Schrödinger equation.

$$i\partial_t|\Psi(t)\rangle = H(t)|\Psi(t)\rangle. \tag{111}$$

The propagator is decomposed as

$$U\left(t,t'\right) = e^{-iK(t)}e^{-iH_{\text{eff}}(t-t')}e^{iK(t')} \tag{112}$$

$$H_{\text{eff}} = \sum_{n=0}^{\infty} H_{\text{eff}}^{(n)} \tag{113}$$

$$H_{\text{eff}}^{(0)} = H_0 \tag{114}$$

$$H_{\text{eff}}^{(1)} = \sum_{m\neq 0} \frac{[H_{-m}, H_m]}{2m\omega} \tag{115}$$

$$H_{\text{eff}}^{(2)} = \sum_{m\neq 0} \frac{[[H_{-m}, H_0], H_m]}{2m^2\omega^2} + \sum_{m\neq 0}\sum_{n\neq 0,m} \frac{[[H_{-m}, H_{m-n}], H_n]}{3mn\omega^2} \tag{116}$$

$$\begin{aligned}
H_{\text{eff}}^{(3)} =& \sum_{m\neq 0} \frac{[[[H_{-m}, H_0], H_0], H_m]}{2m^3\omega^3} + \sum_{m\neq 0}\sum_{n\neq 0,m} \frac{[[[H_{-m}, H_0], H_{m-n}], H_n]}{3m^2n\omega^3} \\
&+ \sum_{m\neq 0}\sum_{n\neq 0,m} \frac{[[[H_{-m}, H_{m-n}], H_0], H_n]}{4mn^2\omega^3} - \sum_{m,n\neq 0} \frac{[[[H_{-m}, H_m], H_{-n}], H_n]}{12mn^2\omega^3} \\
&+ \sum_{m\neq 0}\sum_{n\neq 0,m} \frac{[[H_{-m}, H_0], [H_{m-n}, H_n]]}{12m^2n\omega^3} + \sum_{m,n\neq 0}\sum_{l\neq 0,m,n} \frac{[[[H_{-m}, H_{m-l}], H_{l-n}], H_n]}{6lmn\omega^3} \\
&+ \sum_{m,n\neq 0}\sum_{l\neq 0,m-n} \frac{[[[H_{-m}, H_{m-n-l}], H_l], H_n]}{24lmn\omega^3} + \sum_{m,n\neq 0}\sum_{l\neq 0,m,n} \frac{[[H_{-m}, H_{m-l}], [H_{l-n}, H_n]]}{24lmn\omega^3}
\end{aligned} \tag{117}$$

$$K(t) = \sum_{n=0}^{\infty} K^{(n)}(t) \tag{118}$$

$$iK^{(1)}(t) = -\sum_{m \neq 0} \frac{H_m}{m\omega} e^{-im\omega t} \tag{119}$$

$$iK^{(2)}(t) = \sum_{m \neq 0} \sum_{n \neq 0,m} \frac{[H_n, H_{m-n}]}{2mn\omega^2} e^{-im\omega t} + \sum_{m \neq 0} \frac{[H_m, H_0]}{m^2\omega^2} e^{-im\omega t} \tag{120}$$

$$iK^{(3)}(t) = -\sum_{m \neq 0} \frac{[[H_m, H_0], H_0]}{m^3\omega^3} e^{-im\omega t} + \sum_{m \neq 0} \sum_{n \neq 0} \frac{[H_m, [H_{-n}, H_n]]}{4m^2 n\omega^3} e^{-im\omega t}$$

$$- \sum_{m \neq 0} \sum_{n \neq 0,m} \frac{[[H_n, H_0], H_{m-n}]}{2mn^2\omega^3} e^{-im\omega t} - \sum_{m \neq 0} \sum_{n \neq 0,m} \frac{[[H_n, H_{m-n}], H_0]}{2m^2 n\omega^3} e^{-im\omega t}$$

$$- \sum_{m \neq 0} \sum_{n \neq 0} \sum_{l \neq 0,n,m} \frac{[[H_n, H_{l-n}], H_{m-l}]}{4mnl\omega^3} e^{-im\omega t} - \sum_{m \neq 0} \sum_{n \neq 0} \sum_{l \neq 0,m-n} \frac{[H_n, [H_l, H_{m-n-l}]]}{12mnl\omega^3} e^{-im\omega t}$$

$$\tag{121}$$

## B Commutator calculations

**Lemma 2** *Suppose* $i\mathcal{L}_m\rho = [H_m, \rho]$. *We define for* $N \geq 2$

$$\mathcal{K}_{m_N,\ldots,m_2,m1} = [i\mathcal{L}_{m_N}, [\cdots, [i\mathcal{L}_{m_2}, i\mathcal{L}_{m_1}]]] \tag{122}$$

$$H_{m_N,\ldots,m_2,m1} = [H_{m_N}, [\cdots, [H_{m_2}, H_{m_1}]]]. \tag{123}$$

*Then, it follows that*

$$\mathcal{K}_{m_N,\ldots,m_2,m1}(\rho) = [H_{m_N,\ldots,m_2,m1}, \rho] \tag{124}$$

*for any* $\rho$.

**Proof.** We invoke the mathematical induction. First, for $N = 2$, the statement follows from the Jacobi identity $[A, [B, C]] + [B, [C, A]] + [C, [A, B]] = 0$. In fact,

$$\mathcal{K}_{m_2,m_1}(\rho) = [H_{m_2}, [H_{m_1}, \rho]] - [H_{m_1}, [H_{m_2}, \rho]] \tag{125}$$

$$= [H_{m_2}, [H_{m_1}, \rho]] + [H_{m_1}, [\rho, H_{m_2}]] \tag{126}$$

$$= -[\rho, [H_{m_2}, H_{m_1}]] \tag{127}$$

$$= [H_{m_2,m_1}, \rho]. \tag{128}$$

Next, we suppose that the statement is true for $N = M$ and prove the statement for $N = M + 1$. This step is achieved as follows:

$$\mathcal{K}_{m_{M+1},\ldots,m_2,m1}(\rho) = i\mathcal{L}_{m_{M+1}}\mathcal{K}_{m_M,\ldots,m_2,m1}(\rho) - \mathcal{K}_{m_M,\ldots,m_2,m1}i\mathcal{L}_{m_{M+1}}(\rho) \tag{129}$$

$$= [H_{m_{M+1}}, [H_{m_M,\ldots,m_2,m_1}, \rho]] - [H_{m_M,\ldots,m_2,m_1}, [H_{m_{M+1}}, \rho]] \tag{130}$$

$$= [[H_{m_{M+1}}, H_{m_M,\ldots,m_2,m_1}], \rho] \tag{131}$$

$$= [H_{m_{M+1},\ldots,m_2,m_1}, \rho]. \tag{132}$$

∎

# C   Majorana representation of the XY chain

In Sec. 5.2, we have used the Majorana-fermion representation for the infinite XY chain. Here, we derive the representation and its Fourier transform.

We introduce $2N$ Majorana fermions $w_j$ $(j = 1, 2, \ldots, 2N)$ for an $N$-site chain as

$$w_{2n-1} = \left( \prod_{j=1}^{n-1} \sigma_j^z \right) \sigma_n^x, \tag{133}$$

$$w_{2n} = \left( \prod_{j=1}^{n-1} \sigma_j^z \right) \sigma_n^y, \tag{134}$$

$$w_{2n-1}w_{2n} = i\sigma_n^z, \tag{135}$$

which satisfy the Majorana commutation relations $\{w_i, w_j\} = 2\delta_{ij}$ and translate Eq. (50) into

$$H(t) = -i \sum_{j=1}^{N-1} \left( \frac{1+\gamma}{2} w_{2j}w_{2j+1} - \frac{1-\gamma}{2} w_{2j-1}w_{2(j+1)} \right) - ihf(t) \sum_{j=1}^{N} w_{2j-1}w_{2j}. \tag{136}$$

It is convenient to introduce the two-component fermions $W_j = {}^t(w_{2j-1}, w_{2j})$, which lead to

$$w_{2j-1}w_{2j} = \frac{1}{2}(w_{2j-1}w_{2j} - w_{2j}w_{2j-1}) = {}^tW_j i\tau^y W_j, \tag{137}$$

$$w_{2j}w_{2j+1} = \frac{1}{2}(w_{2j}w_{2j+1} - w_{2j+1}w_{2j}) = \frac{1}{2}{}^tW_j \tau^- W_{j+1} - \frac{1}{2}{}^tW_{j+1}\tau^+ W_j, \tag{138}$$

$$w_{2j-1}w_{2(j+1)} = \frac{1}{2}(w_{2j-1}w_{2(j+1)} - w_{2(j+1)}w_{2j-1}) = \frac{1}{2}{}^tW_j \tau^+ W_{j+1} - \frac{1}{2}{}^tW_{j+1}\tau^- W_j, \tag{139}$$

where $\tau^{\pm} \equiv (\tau^x \pm i\tau^y)/2$ and $\tau^\alpha$ $(\alpha = x, y,$ and $z)$ are the Pauli matrices. These equations give

$$H(t) = -\frac{i}{4}\left[ \sum_{j=1}^{N-1} \left\{ {}^tW_j(\gamma\tau^x - i\tau^y)W_{j+1} - {}^tW_{j+1}(\gamma\tau^x + i\tau^y)W_j \right\} + 2hf(t)\sum_{j=1}^{N} {}^tW_j i\tau^y W_j \right] \tag{140}$$

$$= -\frac{i}{4}{}^t\boldsymbol{W}\boldsymbol{X}\boldsymbol{W}, \tag{141}$$

where $\boldsymbol{W}$ is the $2N$-component vector, and $\boldsymbol{X}$ is the antiHermitian $2N \times 2N$ matrix. Thus, we have obtained the quadratic Majorana representation of the Hamiltonian, which is translation invariant except for the boundaries.

To obtain the quasienergy bands, we derive the time-evolution equation in the Majorana representation. It is convenient to work in the Heisenberg picture to have

$$\frac{d\boldsymbol{W}(t)}{dt} = i[H(t), \boldsymbol{W}(t)] = -\boldsymbol{X}\boldsymbol{W}(t), \tag{142}$$

where $\boldsymbol{W}(t)$ is the Heisenberg-picture operator for $\boldsymbol{W}$ and we have used the commutation relation $\{w_i, w_j\} = 2\delta_{ij}$. To recover the translation symmetry, we consider the infinite chain

$(N \to \infty)$ and invoke the Fourier transform: $W_j \to \widetilde{W}_k \propto \sum_j e^{-ikj} W_j$. In the Fourier basis, we have

$$\frac{dW_k(t)}{dt} = -X(k,t)W_k(t), \tag{143}$$

where

$$X(k) \equiv (\gamma\tau^x - i\tau^y)e^{-ik} - (\gamma\tau^x + i\tau^y)e^{ik} + 2hf(t)i\tau^y = -2i\{\gamma\sin k\tau_x + [\cos k - hf(t)]\tau_y\} \tag{144}$$

is a $2 \times 2$ matrix for each $k$. Physically, the $2 \times 2$ matrix describes the evolution of the quasiparticles.

According to the Floquet theory, the one-cycle evolution

$$V(k) = \exp_+ \left( -\int_0^T X(k,t)dt \right) \tag{145}$$

determines the Floquet modes and the two eigenvalues define the quasienergy bands. Since $X(k,t)$ is antiHermitian and traceless, the eigenvalues of $V(k)$ are given as $e^{\pm i\epsilon(k)T}$, and we regard $\pm\epsilon(k)$ as the quasienergy bands for the Majorana fermions. In the absence of the external drive $f(t) = 0$, we have $\epsilon(k) = 2\sqrt{\cos^2 k + \gamma^2 \sin^2 k}$. Note that $d\epsilon(k)/dk = 0$ at $k = 0$ and $\pm\pi$.

To use the HF expansion for the Hamiltonian, we notice the formal analogy of Eq. (143) to the Schrödinger equation. In fact,

$$h(k,t) \equiv -iX(k,t) = -2\{(\gamma\sin k)\tau^x + [\cos k - hf(t)]\tau^y\} \tag{146}$$

plays the role of Hamiltonian and gives $V(k) = \exp_+[-i\int_0^T h(k,t)dt]$, and, hence, the HF expansion in Appendix A is applicable to $h(k,t)$.

# D   Derivation of time-dependent FLE for weak thermal contact

Historically, dissipative Floquet systems weakly coupled to thermal reservoirs were studied by quantum master equations [41,85], which are equivalent to the FLE. Here, for completeness, we summarize the derivation with slight generalization and emphasis on the Lindblad (GKSL) form. Our derivation follows a textbook [40] in which the GKSL equation is derived for the time-independent $H_S$.

## D.1   Interaction picture

We begin by considering the system-bath composite system whose Hamiltonian reads

$$H(t) = H_S(t) + H_B + H_{SB}, \tag{147}$$

where $H_S(t+T) = H_S(t)$ is for the periodically driven system of interest, $H_B$ for the heat bath (reservoir), and $H_{SB} = \sum_\alpha A_\alpha \otimes B_\alpha$ for the system-bath coupling. Here, precisely speaking, $H_S(t)$ ($H_B$) represents $H_S(t) \otimes 1$ ($1 \otimes H_B$), but we omit $\otimes 1$ and $1\otimes$ for brevity.

Under the Hamiltonian (147), we consider the time evolution of the total system. To this end, it is useful to work in the interaction picture:

$$\frac{\mathrm{d}\rho^I(t)}{\mathrm{d}t} = -\mathrm{i}[H_I(t), \rho^I(t)] \tag{148}$$

and its integral form

$$\rho^I(t) = \rho^I(0) - \mathrm{i} \int_0^t [H_I(s), \rho^I(s)]\mathrm{d}s, \tag{149}$$

where $\rho^I(t)$ is the density matrix for the total system in the interaction picture, and $H_I(t) = U^\dagger(t)H_{SB}U(t)$ with $U(t) = \exp_+\left[-\mathrm{i}\int_0^t (H_S(s) + H_B)\mathrm{d}s\right]$. In the following, we drop the superscript $I$ for density matrices for brevity until we arrive at the final result (179). We assume that, at the initial time $t = 0$, the system and bath are not entangled: $\rho(0) = \rho_S(0) \otimes \rho_B$, where $\rho_B$ represents a static state of the bath. We then substitute Eq. (149) into the RHS of Eq. (148) and take $\mathrm{Tr}_B$ of the both sides, having

$$\frac{\mathrm{d}\rho_S(t)}{\mathrm{d}t} = -\int_0^t ds\mathrm{Tr}_B[H_I(t), [H_I(s), \rho(s)]], \tag{150}$$

where we have used $\mathrm{Tr}_B[\rho(0)B_\alpha] = 0$ for all $\alpha$ without loss of generality. To make this equation Markovian, we make the replacement $\rho(s) \to \rho_S(t) \otimes \rho_B$, by which the RHS only depends on $\rho_S(t)$. We then remove the dependence on the initial time by replacing $s$ in the integral by $t - s$ and extending the integration as $\int_0^t \to \int_0^\infty$. Then, we obtain the following Markovian equation:

$$\frac{d}{dt}\rho_S(t) = -\int_0^\infty ds\mathrm{Tr}_B\left\{[H_I(t), [H_I(t - s), \rho_S(t) \otimes \rho_B]]\right\}, \tag{151}$$

which is Markovian but not in Lindblad form yet. We remark that these approximations are based on physical intuitions, but the errors accompanied by them have recently been quantified in a mathematically rigorous manner [86, 87]. According to the error bounds, the approximations are valid if the system-bath coupling is weak enough and the bath-correlation time is short enough.

We can simplify Eq. (151) by introducing the bath spectral function. To this end, we decompose the system-bath interaction as $H_{SB} = \sum_\alpha A_\alpha \otimes B_\alpha$ and correspondingly

$$H_I(t) = \sum_\alpha A_\alpha(t) \otimes B_\alpha(t), \tag{152}$$

where $A_\alpha(t) = U_S^\dagger(t)A_\alpha U_S(t)$ with $U_S(t) = \exp_+\left(-\mathrm{i}\int_0^t ds H_S(s)\right)$, and $B_\alpha(t) = \mathrm{e}^{\mathrm{i}H_B t}B_\alpha \mathrm{e}^{-\mathrm{i}H_B t}$. We define the bath spectral function by

$$\langle B_\alpha(t)B_\beta(t - s)\rangle \equiv \mathrm{Tr}_B\left(\rho_B B_\alpha(t)B_\beta(t - s)\right). \tag{153}$$

Although the above arguments apply to any $\rho_B$, we here assume $\rho_B = \mathrm{e}^{-\beta H_B}/Z_B$. Then we have $\langle B_\alpha(t)B_\beta(t - s)\rangle = \langle B_\alpha(s)B_\beta(0)\rangle$ and the so-called Kubo-Martin-Schwinger (KMS)

condition: $\langle B_\alpha(t)B_\beta(0)\rangle = \langle B_\beta(s)B_\alpha(t+\mathrm{i}\beta)\rangle$. By using the bath spectral function, Eq. (151) is rewritten as

$$\frac{d}{dt}\rho_S(t) = \sum_{\alpha,\beta}\int_0^\infty ds\langle B_\alpha(s)B_\beta(0)\rangle\left[A_\beta(t-s)\rho_S(t), A_\alpha(t)\right] + \mathrm{H.c.} \tag{154}$$

Equation (154) can be further simplified once we Fourier expand $A_\alpha(t)$ and $A_\beta(t-s)$. For the conventional setup where $H_S$ is time-independent, the Fourier expansion is done simply by the decomposition into the eigenmodes of $H_S$. However, in our periodically driven systems, the decomposition into the Floquet states [6,7] are useful:

$$i\frac{d}{dt}|\psi_m(t)\rangle = H_S(t)|\psi_m(t)\rangle; \qquad |\psi_m(t)\rangle = e^{-\mathrm{i}\epsilon_m t}|u_m(t)\rangle; \qquad |u_m(t+T)\rangle = |u_m(t)\rangle. \tag{155}$$

Here, $\{|\psi_m(t)\rangle\}_{m=1}^N$ ($N < \infty$ is the Hilbert-space dimension) are independent solutions for the time-dependent Schrödinger equation without system-bath coupling, and we call $\epsilon_m$ and $|u_m(t)\rangle$ the quasienergy and Floquet state, respectively. We remark that $\epsilon_m$ and $|u_m(t)\rangle$ are not uniquely defined, but we can redefine them by $\epsilon_m \to \epsilon_m + k\omega$ and $|u_m(t)\rangle \to e^{i\epsilon_m t}|u_m(t)\rangle$ without changing $|\psi_m(t)\rangle$ and the periodicity of $|u_m(t)\rangle$. For concrete calculations, one may fix a set of $\epsilon_m$'s, but the physical observables must be invariant under those shifts.

These states give

$$U_S(t) = \sum_{m=1}^N |\psi_m(t)\rangle\langle\psi_m(0)|, \tag{156}$$

which leads to

$$A_\alpha(t) = \sum_{m,n}|\psi_m(0)\rangle\langle\psi_m(t)|A_\alpha|\psi_n(t)\rangle\langle\psi_n(0)|. \tag{157}$$

We note that $\langle\psi_m(t)|A_\alpha|\psi_n(t)\rangle = e^{\mathrm{i}(\epsilon_m-\epsilon_n)t}\langle u_m(t)|A_\alpha|u_n(t)\rangle$ has a discrete spectrum, in which nonzero weights lie at $\epsilon = -(\epsilon_m-\epsilon_n)+\ell\omega$ ($\ell \in \mathbb{Z}$), since the Floquet states $|u_m(t)\rangle$ are periodic. Thus we introduce the spectral decomposition for the matrix elements as

$$\langle\psi_m(t)|A_\alpha|\psi_n(t)\rangle = \sum_\epsilon \mathsf{A}_{mn}^\alpha(\epsilon)e^{-\mathrm{i}\epsilon t} = \sum_{\ell\in\mathbb{Z}}\mathsf{A}_{mn}^\alpha(\epsilon_n-\epsilon_m+\ell\omega)e^{-\mathrm{i}(\epsilon_n-\epsilon_m+\ell\omega)t} \tag{158}$$

and correspondingly that for the operators as

$$A_\alpha(t) = \sum_\epsilon e^{-\mathrm{i}\epsilon t}A_\epsilon^\alpha; \qquad A_\epsilon^\alpha \equiv \sum_{m,n}\mathsf{A}_{mn}^\alpha(\epsilon)|\psi_m(0)\rangle\langle\psi_n(0)|. \tag{159}$$

For later use, let us show that $A_\epsilon^\alpha$ and $(A_\epsilon^\alpha)^\dagger$ are operators that lower and raise quasienergy by $\epsilon$:

$$U_F A_\epsilon^\alpha U_F^\dagger = e^{+\mathrm{i}\epsilon T}A_\epsilon^\alpha, \qquad U_F A_\epsilon^{\alpha\dagger}U_F^\dagger = e^{-\mathrm{i}\epsilon T}A_\epsilon^{\alpha\dagger}, \tag{160}$$

where

$$U_F \equiv U_S(T) = \sum_{m=1}^N |\psi_m(T)\rangle\langle\psi_m(0)| = \sum_{m=1}^N e^{-\mathrm{i}\epsilon_m T}|\psi_m(0)\rangle\langle\psi_m(0)| \tag{161}$$

denotes the one-cycle unitary evolution. To prove that, we first notice that

$$U_F |\psi_m(0)\rangle = |\psi_m(T)\rangle = e^{-i\epsilon_m T} |\psi_m(0)\rangle , \tag{162}$$

meaning that $|\psi_m(0)\rangle$ is an eigenstate of $U_F$ with eigenvalue $e^{-i\epsilon_m T}$. This equation also implies

$$\langle\psi_n(0)| U_F^\dagger = e^{i\epsilon_n T} \langle\psi_n(0)| . \tag{163}$$

Using Eqs. (162) and (163) together with Eq. (159), we have

$$U_F A_\epsilon^\alpha U_F^\dagger = \sum_{m,n} \mathsf{A}_{mn}^\alpha(\epsilon) e^{i(\epsilon_n - \epsilon_m)T} |\psi_m(0)\rangle \langle\psi_n(0)| \tag{164}$$

$$= \sum_{m,n} \mathsf{A}_{mn}^\alpha(\epsilon) e^{i\epsilon T} |\psi_m(0)\rangle \langle\psi_n(0)| \tag{165}$$

$$= e^{i\epsilon T} A_\epsilon^\alpha . \tag{166}$$

Here, to obtain the second line, we have used the fact that $\mathsf{A}_{mn}^\alpha(\epsilon)$ is nonvanishing only when $\epsilon_n - \epsilon_m = \epsilon + \ell\omega$ for some $\ell \in \mathbb{Z}$. Equation (166) is the first equation in Eq. (160) that we wanted to prove, and we obtain the second one by taking the Hermitian conjugation of the first one.

To show this, we recall the following characterization of the quasienergy:

$$U_S(T) |\psi_n(0)\rangle = \mathrm{e}^{-i\epsilon_n T} |\psi_n(0)\rangle , \tag{167}$$

where each quasienergy $\epsilon_n$ is defined modulo $\omega = 2\pi/T$. Now, we consider the Floquet eigenstate multiplied by $A_\epsilon^\alpha$

$$A_\epsilon^\alpha |\psi_n(0)\rangle = \sum_m \mathsf{A}_{mn}^\alpha(\epsilon) |\psi_m(0)\rangle . \tag{168}$$

To look into its quasienergy, we multiply $U_S(T)$ onto it, having

$$U_S(T)[A_\epsilon^\alpha |\psi_n(0)\rangle] = \sum_m \mathsf{A}_{mn}^\alpha(\epsilon)\mathrm{e}^{-i\epsilon_m T} |\psi_m(0)\rangle . \tag{169}$$

Here we remember that $\mathsf{A}_{mn}^\alpha(\epsilon)$ is nonvanishing only for pairs $(m,n)$ such that $\epsilon_n - \epsilon_m \equiv \epsilon$ ( mod $\omega$). Therefore $\mathrm{e}^{-i\epsilon_m T}$ in the sum can be replaced by $\mathrm{e}^{-i(\epsilon_n-\epsilon)T}$, which can be put out of the sum:

$$U_S(T)[A_\epsilon^\alpha |\psi_n(0)\rangle] = \mathrm{e}^{-i(\epsilon_n-\epsilon)T} \sum_m \mathsf{A}_{mn}^\alpha(\epsilon) |\psi_m(0)\rangle = \mathrm{e}^{-i(\epsilon_n-\epsilon)T}[A_\epsilon^\alpha |\psi_n(0)\rangle]. \tag{170}$$

This equation means that $A_\epsilon^\alpha |\psi_n(0)\rangle$ is a Floquet eigenstate with quasienergy $\epsilon_n - \epsilon$ and that $A_\epsilon^\alpha$ lowers the quasienergy by $\epsilon$.

Finally, we summarize the above results as operator relations ($U_F \equiv U_S(T)$):

$$U_F A_\epsilon^\alpha U_F^\dagger = \mathrm{e}^{+i\epsilon T} A_\epsilon^\alpha, \tag{171}$$

$$U_F A_\epsilon^{\alpha\dagger} U_F^\dagger = \mathrm{e}^{-i\epsilon T} A_\epsilon^{\alpha\dagger}. \tag{172}$$

A proof of Eq. (171) is as follows:

$$U_F A_\epsilon^\alpha U_F^\dagger = \sum_{m,n} \mathsf{A}_{mn}^\alpha(\epsilon) |\psi_m(T)\rangle \langle\psi_n(T)| \tag{173}$$

$$= \sum_{m,n} \mathsf{A}_{mn}^\alpha(\epsilon) e^{-i(\epsilon_m - \epsilon_n)T} |\psi_m(0)\rangle \langle\psi_n(0)| \tag{174}$$

$$= \sum_{m,n} \mathsf{A}_{mn}^\alpha(\epsilon) e^{+i\epsilon T} |\psi_m(0)\rangle \langle\psi_n(0)| = e^{+i\epsilon T} A_\epsilon^\alpha. \tag{175}$$

Here, from the second to third lines, we have used the fact that $\mathsf{A}_{mn}^\alpha(\epsilon)$ is nonvanishing only for pairs $(m,n)$ such that $\epsilon_n - \epsilon_m \equiv \epsilon \ (\mod \omega)$. Equation (172) follows from Eq. (171).

We are ready to obtain the Lindblad form with the final approximation, the rotating wave approximation (RWA). Substituting Eq. (159) into Eq. (154), we have

$$\frac{d}{dt}\rho_S(t) = \left( \sum_{\alpha,\beta,\epsilon,\epsilon'} \Gamma_{\alpha\beta}(\epsilon) e^{i(\epsilon-\epsilon')t} \left[ A_\epsilon^\beta \rho_S(t) A_{\epsilon'}^{\alpha\dagger} - A_\epsilon^{\alpha\dagger} A_{\epsilon'}^\beta \rho_S(t) \right] \right) + \text{H.c.}, \tag{176}$$

where we have defined the bath spectral function

$$\Gamma_{\alpha\beta}(\epsilon) = \int_0^\infty ds\, e^{i\epsilon s} \langle B_\alpha(s) B_\beta(0)\rangle. \tag{177}$$

Since $\Gamma_{\alpha\beta}(\epsilon)$ is not a Hermitian matrix, it is convenient to decompose it into the Hermitian and antiHermitian parts:

$$\Gamma_{\alpha\beta}(\epsilon) = \frac{1}{2}\gamma_{\alpha\beta}(\epsilon) + iS_{\alpha\beta}(\epsilon). \tag{178}$$

The RWA means that we keep $\epsilon = \epsilon'$ terms only in Eq. (176), by which we arrive at the following equation of Lindblad form:

$$\frac{d}{dt}\rho_S^I(t) = -i[\Lambda^{\text{LS}}, \rho_S^I(t)] + \mathcal{D}^I(\rho_S^I(t)) \tag{179}$$

where the Lamb shift $\Lambda^{\text{LS}}$ and dissipator $\mathcal{D}^I$ are defined by

$$\Lambda^{\text{LS}} = \sum_{\alpha,\beta,\epsilon} S_{\alpha\beta}(\epsilon) A_\epsilon^{\alpha\dagger} A_\epsilon^\beta, \tag{180}$$

$$\mathcal{D}^I = \sum_{\alpha,\beta,\epsilon} \gamma_{\alpha\beta}(\epsilon) \left[ A_\epsilon^\beta \rho_S^I(t) A_\epsilon^{\alpha\dagger} - \frac{1}{2}\left\{ A_\epsilon^{\alpha\dagger} A_\epsilon^\beta, \rho_S^I(t) \right\} \right], \tag{181}$$

Here we have recovered the superscript $I$ on $\rho_S(t)$ to emphasize that this is in the interaction picture.

The RWA is valid when quasienergy spacings are large enough but becomes invalid if the quasienergy spacings become infinitely small with the system-bath coupling and the bath correlation time being fixed. In this sense, Eq. (179) cannot apply to large nonintegrable many-body systems, where (quasi)energy spacings decrease exponentially in the system size. Thus, Eq. (179) should apply, e.g., to two- or few-level quantum systems like an atom, small clusters of spins, or some integrable quantum systems with nonvanishing energy gaps. There

have been new approaches to derive master equations of Lindblad form without using the RWA [86, 87].

We note that our derivation for Eq. (179) is slightly generalized from that of the previous studies [41, 85] in the following sense. We recall that $\epsilon$ takes the values in $\{\Delta_{mn}(\ell) \equiv \epsilon_n - \epsilon_m + \ell\omega\}_{m,n,k}$. Here, we emphasize that the correspondence between $\epsilon$ and $(m, n, \ell)$ is not one-to-one in general, and there are multiple $(m, n, \ell)$'s that give the same $\epsilon$. This situation occurs, for example, in the following three cases:

1. The quasienergies $\{\epsilon_m\}$ are degenerate.

2. The quasienergy differences $\{\epsilon_m - \epsilon_n\}$ are degenerate, such as for equidistant $\{\epsilon_m\}$.

3. There are some pair $(m, m')$ such that $\epsilon_m - \epsilon_{m'} = \omega/2$.

Our derivation of Eq. (179) is valid even if the correspondence between $\epsilon$ and $(m, n, \ell)$ is not one-to-one in contrast to that in the previous studies [41, 85].

We note that the Lamb shift $\Lambda^{\mathrm{LS}}$ does not change the quasienergy. This follows from the above-shown fact that $A_\epsilon^{\alpha\dagger}$ ($A_\epsilon^\beta$) raises (lowers) the quasienergy by $\epsilon$. As an operator expression, we have

$$[U_F, \Lambda^{\mathrm{LS}}] = 0, \tag{182}$$

which follows from Eq. (160). This is a Floquet counterpart of the following property: The Lamb shift does not change the energy of the system in the time-independent case. As a consequence, when the quasienergies are not degenerate, $\Lambda^{\mathrm{LS}}$ is diagonal in the Floquet eigenbasis

$$\Lambda^{\mathrm{LS}} = \sum_m \lambda_m |\psi_m(0)\rangle \langle\psi_m(0)| \tag{183}$$

and hence only shifts the quasienergy by $\lambda_m$.

### D.2    Schrödinger picture

Equation (179) is written in the interaction picture. To go back to the Schrödinger picture, we remember $\rho_S(t) = U_S(t)\rho_S^I(t)U_S^\dagger(t)$. Then we have

$$\frac{d}{dt}\rho_S(t) = -\mathrm{i}[H_S(t) + \Lambda^{\mathrm{LS}}(t), \rho_S(t)] + \sum_{\alpha,\beta,\epsilon} \gamma_{\alpha\beta}(\epsilon)\left[A_\epsilon^\beta(t)\rho_S(t)A_\epsilon^{\alpha\dagger}(t) - \frac{1}{2}\left\{A_\epsilon^{\alpha\dagger}(t)A_\epsilon^\beta(t), \rho_S(t)\right\}\right].$$
$$\tag{184}$$

Here the jump operators are given by

$$A_\epsilon^\alpha(t) = e^{-\mathrm{i}\epsilon t}\sum_{m,n} \mathsf{A}_{mn}^\alpha(\epsilon)|\psi_m(t)\rangle\langle\psi_n(t)|, \tag{185}$$

and the Lamb shift is by

$$\Lambda^{\mathrm{LS}}(t) = \sum_{\alpha,\beta,\epsilon} S_{\alpha\beta}(\epsilon)A_\epsilon^{\alpha\dagger}(t)A_\epsilon^\beta(t). \tag{186}$$

Here, in $A_\epsilon^\alpha(t)$, we put the oscillating phase factor $e^{-i\epsilon t}$ so that $A_\epsilon^\alpha(t)$ becomes time-periodic (this phase factor keeps the Lindbladian invariant as $A_\epsilon^{\alpha\dagger}(t)$ and $A_\epsilon^\beta$ appear in pairs). To show the periodicity $A_\epsilon^\alpha(t) = A_\epsilon^\alpha(t + T)$, we write Eq. (185) in terms of $|u_m(t)\rangle$: $A_\epsilon^\alpha(t) = \sum_{m,n} \mathsf{A}_{mn}^\alpha(\epsilon) e^{i(-\epsilon-\epsilon_m+\epsilon_n)t} |u_m(t)\rangle \langle u_n(t)|$, in which we need to prove $e^{i(-\epsilon-\epsilon_m+\epsilon_n)t}$ is periodic ($|u_m(t)\rangle$ are periodic by definition). Since $\mathsf{A}_{mn}^\alpha(\epsilon)$ is nonvanishing only when $\epsilon = \epsilon_n - \epsilon_m + k\omega$ for integers $k$, the oscillating phase factor $e^{i(-\epsilon-\epsilon_m+\epsilon_n)t}$ becomes $e^{-ik\omega t}$, which is periodic.

It is convenient to introduce the dissipator $\mathcal{D}_t$, which is the following superoperator

$$\mathcal{D}_t(\rho) \equiv \sum_{\alpha,\beta,\epsilon} \gamma_{\alpha\beta}(\epsilon) \left[ A_\epsilon^\beta(t) \rho A_\epsilon^{\alpha\dagger}(t) - \frac{1}{2} \left\{ A_\epsilon^{\alpha\dagger}(t) A_\epsilon^\beta(t), \rho \right\} \right]. \tag{187}$$

We also introduce the Lindbladian superoperator by

$$\mathcal{L}_t(\rho) = -i[H_S(t) + \Lambda^{\mathrm{LS}}(t), \rho] + \mathcal{D}_t(\rho). \tag{188}$$

One can easily confirm that this FLE reduces to the well-known Lindblad equation for undriven systems [40], noting that, in undriven systems, each Floquet eigenstate $|\psi_m(t)\rangle$ corresponds to the time-evolving energy eigenstate $e^{-iE_m t} |E_m\rangle$ for $H^S |E_m\rangle = E_m |E_m\rangle$.

It is noteworthy that the Lindbladian of the FLE is time-independent in the interaction picture (179) while not in the Schrödinger picture (184). This is an extraordinary property of the FLE based on the weak system-bath coupling and the RWA, and we cannot find such a nice frame in which the time dependence is eliminated for a general FLE in the Schrödinger picture. In this sense, the FLE derived in this Appendix is special, making analysis easier.

## D.3   Time evolution of density matrix

Here we analyze the time evolution of density matrices, showing that their off-diagonal elements decay whereas the diagonal ones obey the master equation. In Ref. [85], these properties were obtained under the strong assumption: If $\epsilon_n - \epsilon_m + \ell\omega = \epsilon_{n'} - \epsilon_{m'} + \ell'\omega$, then $(m, n, \ell) = (m', n', \ell')$ holds true. Our argument here relaxes this assumption and also applies when the quasienergies are nondegenerate as long as the NESS is unique. We also discuss how the results change for degenerate quasienergies.

### D.3.1   Nondegenerate quasienergies

We first discuss the case when the quasienergies are not degenerate. To analyze the time evolution, it is convenient to work in the interaction picture (179) and represent the density matrix in the Floquet-state basis:

$$\rho_S^I(t) = \sum_{m,n} \sigma_{mn}(t) |m\rangle \langle n|, \tag{189}$$

where we have introduced the abbreviation $|m\rangle = |\psi_m(0)\rangle$. Substituting Eq. (189) into Eq. (179), we have

$$\frac{d\sigma_{mn}(t)}{dt} = -i(\lambda_m - \lambda_n)\sigma_{mn}(t) + \sum_{\alpha,\beta,\epsilon} \gamma_{\alpha\beta}(\epsilon) \left[ \sum_{m',n'} \langle m|A_\epsilon^\beta|m'\rangle \sigma_{m'n'}(t) \langle n'|A_\epsilon^{\alpha\dagger}|n\rangle \right.$$

$$\left. - \frac{1}{2} \sum_{m'} \langle m|A_\epsilon^{\alpha\dagger} A_\epsilon^\beta|m'\rangle \sigma_{m'n}(t) - \frac{1}{2} \sum_{n'} \sigma_{mn'}(t) \langle n'|A_\epsilon^{\alpha\dagger} A_\epsilon^\beta|n\rangle \right]. \tag{190}$$

To simplify the sums in Eq. (190), we recall that $A_\epsilon^\beta$ and $(A_\epsilon^\alpha)^\dagger$ are operators that lower and raise quasienergy by $\epsilon$ as shown in Eq. (160). Thus, $A_\epsilon^{\alpha\dagger} A_\epsilon^\beta$ does not raise or lower quasienegies, meaning that $\langle m|A_\epsilon^{\alpha\dagger} A_\epsilon^\beta|m'\rangle \propto \delta_{mm'}$ as the quasienergies are not degenerate. Together with Eq. (88), we can rewrite Eq. (190) as

$$
\frac{d\sigma_{mn}(t)}{dt} = -i(\lambda_m - \lambda_n)\sigma_{mn}(t) - \frac{1}{2}\sum_{n'}(W_{n'm} + W_{n'n})\,\sigma_{mn}(t)
$$
$$
+ \sum_{\alpha,\beta,\epsilon}\gamma_{\alpha\beta}(\epsilon)\sum_{m',n'}\langle m|A_\epsilon^\beta|m'\rangle\,\sigma_{m'n'}(t)\,\langle n'|A_\epsilon^{\alpha\dagger}|n\rangle. \tag{191}
$$

We now have a set of $N^2$ differential equations for $1 \le m, n \le N$, part of which are coupled by the final term on the right-hand side (RHS) of Eq. (191).

Here, we can prove the remarkable property of this set of equations that the diagonal ones ($\{\sigma_{mm}(t)\}_{m=1}^N$) and off-diagonal ones ($\{\sigma_{mn}(t)\}_{m\neq n}$ are decoupled with each other. To show this, we first substitute $n$ by $m$ in Eq. (191) and prove that only $m' = n'$ terms appear on the RHS. As $A_\epsilon^\beta$ ($A_\epsilon^{\alpha\dagger}$) lowers (raises) the quasienergy by $\epsilon$, $\sigma_{m'n'}(t)$ appears only when $\epsilon_m \equiv \epsilon_{m'} - \epsilon$ and $\epsilon_m \equiv \epsilon_{n'} - \epsilon \mod \omega$, implying $m' = n'$. Thus, we have obtained the result that the diagonal elements $\{\sigma_{mm}(t)\}_m$ form a closed set of equations. Second, we suppose $m \neq n$ in Eq. (191) and prove that only $m' \neq n'$ terms appear on the RHS. The proof goes similarly. One can easily check that the set of equations for the diagonal elements are the same as Eq. (87).

The off-diagonal elements $\sigma_{m\neq n}(t)$ form a complex set of equations except for the special case studied previously [85], where $\epsilon_n - \epsilon_m + \ell\omega = \epsilon_{n'} - \epsilon_{m'} + \ell'\omega$ means $m = m'$, $n = n'$, and $\ell = \ell'$. In this special case, the final term on the RHS of Eq. (191) vanishes for $m \neq n$, and each $\sigma_{m\neq n}(t)$ obeys a closed differential equation. As $W_{n'm} + W_{n'n} > 0$, the equation means that each $\sigma_{m\neq n}(t)$ exponentially decreases with oscillation to vanish as $t \to \infty$. However, this special case does not necessarily occur even if we assume that the quasienergies are not degenerate (recall the example cases 2 and 3 discussed in Sec. D.1.

Nevertheless, in the weaker assumption of nondegenerate quasienergies, we can prove that the off-diagonal elements $\sigma_{m\neq n}$ vanish in the NESS as long as the NESS is unique. For this purpose, we invoke a symmetry argument in the Lindbladian systems [106–108]. Let us symbolically represent the interaction picture (179) as

$$
\frac{d}{dt}\rho_S^I(t) = \mathcal{L}_I[\rho_S^I(t)]. \tag{192}
$$

Now we notice that the Lindbladian $\mathcal{L}_I$ has the following weak symmetry [106] associated with the quasienergy:

$$
[\mathcal{L}_I, \mathcal{U}_F] = 0, \tag{193}
$$

where $\mathcal{U}_F$ is a superoperator defined by $\mathcal{U}_F(\rho) \equiv U_F \rho U_F^\dagger$. One can easily confirm Eq (193) by using Eqs. (160) and (182). One important consequence of this weak symmetry is that there exists a stationary solution of the following form [106]: $\rho = \sum_\mu p_\mu \tau_\mu$, where $\{\tau_\mu\}$ are matrices belonging to the zero eigenvalue of $\mathcal{U}$. For our $\mathcal{U}$ ($\mathcal{U}\rho \equiv U_F \rho U_F^\dagger$), these zero-eigenvalue states are $\tau_\mu = |n\rangle\langle n|$. As we have assumed that the NESS is unique, this type of stationary solution is the NESS, implying that $\sigma_{m\neq n}(t) \to 0$ as $t \to \infty$.

### D.3.2 Degenerate quasienergies

When the quasienergies $\epsilon_m$'s are degenerate, we cannot, in general, have the decoupling of the diagonal and off-diagonal elements $\sigma_{mn}(t)$ of the density matrix. This is due to, e.g., $\langle m | A_\epsilon^{\alpha\dagger} A_\epsilon^\beta | m' \rangle \not\propto \delta_{mm'}$.

Yet, the weak symmetry constrains the form of the NESS density matrix. To see this, we introduce a new index $a$ that distinguishes the degenerate Floquet states. In this new notation, Eq. (161) is represented as

$$U_F = \sum_{m=1}^{d} e^{-i\epsilon_m T} \sum_{a=1}^{N_m} |m, a\rangle \langle m, a|, \tag{194}$$

where $d$ is the number of distinct quasienergies, $N_m$ the degree of degeneracy of each quasienergy $\epsilon_m$, and $|m, a\rangle = |\psi_{m,a}(0)\rangle$. Thus, the zero-eigenvalue states for the superoperator $\mathcal{U}_F(\rho) = U_F \rho U_F^\dagger$ are $|m, a\rangle \langle m, b|$, and the weak symmetry tells us that the NESS is represented as

$$\rho_{\text{ness}}^I = \sum_{m=1}^{d} \sum_{a,b=1}^{N_m} c_{ab}^m |m, a\rangle \langle m, b|. \tag{195}$$

The expansion coefficients within each degenerate subspace $c_{ab}^m$ depend on models. As $c_{ab}^m$ are Hermitian matrices for each $m$, they are diagonalizable by a unitary basis transformation, so is $\rho_{\text{ness}}^I$. Note, however, that the differential equations for $\sigma_{mn}(t)$ cannot, even in this basis, be separated for the diagonal and off-diagonal elements like in the nondegenerate case.

### D.4 Conditions for FGS when quasienergies are degenerate

Here we show that, within the FLE obtained by the RWA, the NESS coincides with the FGS for an extremely large frequency even if the quasienergies are degenerate. In this subsection, we work in the interaction picture (179) and discuss when $\rho_{\text{ness}}^I$ sufficiently approaches $\rho_{\text{FG}}^I = e^{-\beta H_F}/Z$, where

$$H_F = \sum_m \epsilon_m |\psi_m(0)\rangle \langle \psi_m(0)|. \tag{196}$$

As discussed in Sec. 6.1.1, the quasienergies $\epsilon_m$'s are defined only modulo $\omega$, and the FGS is ill-defined in general. Here we consider the situation that $\omega$ is much larger than the system's energy scale, i.e., $H_F$ becomes close enough to $H_0$ ($H_0$ is the undriven Hamiltonian). In such a case, we can uniquely determine $\epsilon_m$'s so that $\epsilon_m - E_m = O(1/\omega)$ holds for each $m$ ($E_m$ are the eigenenergies of $H_0$). Also, we assume that $\omega$ is so large that

$$|\epsilon_m - \epsilon_n| < \omega \qquad \forall m, n. \tag{197}$$

To judge if $\rho_{\text{FG}}^I = e^{-\beta H_F}/Z$ is a steady-state solution we substitute it into the right-hand side of the FLE (179) and ask if it vanishes. Assuming the basis on which $\Lambda^{\text{LS}}$ is diagonal (183), we have $[\Lambda^{\text{LS}}, \rho_{\text{FG}}^I] = 0$. Thus, we are to show $\mathcal{D}^I(\rho_{\text{FG}}^I) = 0$.

First, let us argue that $\mathcal{D}^I(\rho_{\text{FG}}^I) = 0$ holds if we are allowed to use

$$[H_F, A_\epsilon^\alpha] = -\epsilon A_\epsilon^\alpha, \qquad [H_F, A_\epsilon^{\alpha\dagger}] = \epsilon A_\epsilon^{\alpha\dagger}, \tag{198}$$

which are stronger versions of the commutation relations (160). Assuming Eq. (198) as a working hypothesis, we have

$$\mathcal{D}^I(\rho^I_{\mathrm{FG}}) = Z^{-1} \sum_{\alpha,\beta,\epsilon} \gamma_{\alpha\beta}(\epsilon) \left[ A^\beta_\epsilon e^{-\beta H_F} A^{\alpha\dagger}_\epsilon - \frac{1}{2} \left\{ A^{\alpha\dagger}_\epsilon A^\beta_\epsilon, e^{-\beta H_F} \right\} \right] \tag{199}$$

$$= Z^{-1} e^{-\beta H_F} \sum_{\alpha,\beta,\epsilon} \left[ e^{-\beta\epsilon} \gamma_{\alpha\beta}(\epsilon) A^\beta_\epsilon A^{\alpha\dagger}_\epsilon - \gamma_{\alpha\beta}(\epsilon) A^{\alpha\dagger}_\epsilon A^\beta_\epsilon \right], \tag{200}$$

where we have used $[H_F, A^{\alpha\dagger}_\epsilon A^\beta_\epsilon] = 0$ and $A^\beta_\epsilon e^{-\beta H_F} = e^{-\beta\epsilon} e^{-\beta H_F} A^\beta_\epsilon$, which follow from our working hypothesis (198) and the Baker–Campbell–Hausdorff formula. Then we rewrite the first term in the sum of Eq. (200) by changing the dummy index as $\epsilon \to -\epsilon$, having

$$\sum_{\alpha,\beta,\epsilon} e^{-\beta\epsilon} \gamma_{\alpha\beta}(\epsilon) A^\beta_\epsilon A^{\alpha\dagger}_\epsilon = \sum_{\alpha,\beta,\epsilon} e^{\beta\epsilon} \gamma_{\alpha\beta}(-\epsilon) A^\beta_{-\epsilon} A^{\alpha\dagger}_{-\epsilon} \tag{201}$$

$$= \sum_{\alpha,\beta,\epsilon} e^{\beta\epsilon} \gamma_{\alpha\beta}(-\epsilon) A^\beta_{\epsilon\dagger} A^\alpha_\epsilon \tag{202}$$

$$= \sum_{\alpha,\beta,\epsilon} \gamma_{\beta\alpha}(\epsilon) A^{\beta\dagger}_\epsilon A^\alpha_\epsilon, \tag{203}$$

where we have used the KMS condition (76) and $A^\alpha_{-\epsilon} = (A^\alpha_\epsilon)^\dagger$. Substituting Eq. (203) into Eq. (200) and changing the dummy indices appropriately, we obtain

$$\mathcal{D}^I(\rho^I_{\mathrm{FG}}) = 0. \tag{204}$$

Thus, we have shown that the FGS is a steady-state if our working hypothesis (198) are justified.

Finally, we justify Eq. (198) when the driving frequency is much larger than the bath spectral cutoff. From the definition of $A^\alpha_\epsilon$ (81), we have

$$[H_F, A^\alpha_\epsilon] = \sum_{m,n} (\epsilon_m - \epsilon_n) \mathsf{A}^\alpha_{mn}(\epsilon) |\psi_m(0)\rangle \langle\psi_n(0)|. \tag{205}$$

We recall that $\mathsf{A}^\alpha_{mn}(\epsilon)$ is nonvanishing only when $\epsilon = \epsilon_n - \epsilon_m + k\omega$ for integer $k$'s. However, like in the argument in Sec. 6.1.2, the $k \neq 0$ contributions are vanishingly small because it is accompanied by the bath spectral function $\gamma_{\alpha\beta}(\epsilon) = \gamma_{\alpha\beta}(\epsilon_n - \epsilon_m + k\omega) \approx 0$ if $\omega \gg \Lambda$. Therefore, we can neglect $k \neq 0$ contributions and set $\epsilon_m - \epsilon_n = -\epsilon$ in Eq. (205), which leads to

$$[H_F, A^\alpha_\epsilon] = \sum_{m,n} (-\epsilon) \mathsf{A}^\alpha_{mn}(\epsilon) |\psi_m(0)\rangle \langle\psi_n(0)| = -\epsilon A^\alpha_\epsilon. \tag{206}$$

Thus, we have shown that Eqs. (198) are justified when the driving frequency is much larger than the bath spectral cutoff and hence $\rho^I_{\mathrm{ness}} \approx \rho^I_{\mathrm{FG}}$.

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
