# Peer review of "Nonequilibrium steady states in the Floquet-Lindblad systems: van Vleck's high-frequency expansion approach"

_SciPost Physics Core_

## Round 2 · Referee Report · Anonymous (Referee 1) · 2021-9-3

Strengths

1 - The Floquet-Lindblad equation analysed in this work is important for many systems of current interest. The inclusion of dissipation is relevant for open quantum systems and solid state systems.

2 - The expansion allows to calculate the NESS at high frequency just by looking at the spectrum of the effective Liouvillian $\mathcal{L}_{\textrm{eff}}$, which can be systematically computed by way of the expansion. No study of the transient to the NESS is required.

3 - The properties of the expansion and of the NESS are clearly presented. The effectiveness of the method is shown for several model.

4 - The state of the art in the field is properly addressed and the relevant literature is well referenced.

5 - In the case of time dependent dissipators, it is shown that the NESS coincide with the "quasi-thermal" Floquet-Gibbs state when the spectral width of the bath is much smaller than the drive frequency. This can have important applications to electronic systems driven with high frequency lasers.

Weaknesses

1 - The high $\omega$ expansion does not seem to be easily tractable analytically, in general. The authors do not present a fully analytic study of a sample system: for the simplest example (XY chain) the effects of dissipation on the phase diagram are treated qualitatively.

2 - It is not clear wether a general rule exists to quantify the error of the expansion, similarly to Eq. (66).

3 - The conclusions present a limited outlook view.

4 - Figure 3, 4 and 6 are not very legible.

Report

In this paper the authors present a systematic high-frequency expansion of the quantum master equation to study the non-equilibrium steady state of a Floquet-Lindblad system subject to periodic drives and to dissipation, and apply it to selected models.

The topic is of current relevance, given the advances in the study of driven systems, as properly shown in the cited literature. The study of dissipation is also important for many solid-state systems where the driven degrees of freedom are coupled to dissipative baths of various kinds.

The authors review the most common techniques used to study a NESS, and nicely present the generalization of the Van Vleck expansion (Ref. 8 and 9) to dissipative Lindblad equations. The properties of the equations and of the NESS are clearly stated.
The authors then apply their findings to an XY chain, a three level system modelling NV centres, and to the inverse Faraday effect, study the properties of the NESS and the applicability of the expansion.

The article is mostly well written and orderly presented, especially in Sec. 1-4. The properties of the expansion are nicely stated in section 5.1 and 6.1, but the presentation of the results for model systems is less organized. The appendices are very useful in understanding the main text.

I have a few remarks.

a) The high-expansion analysis of Sec. 5.2 is performed on the hamiltonian evolving $W_k$ (Majorana fermions), and it is not explained how to relate it to a high-frequency expansion for the spin density matrix.

b) The phase diagram in Fig. 1 changes drastically from 0th to 2nd order expansion. Is an expansion to 2nd order sufficient to get the full picture of the physics involved?

c) It is not immediately clear to me why Eq. (50) implies property (ii).

d) In Eqs. (51)-(55), the dissipator $\mathcal D$ commutes with $\mathcal S_z$ independently of the sites on which the jump operators act. Therefore, while it is true that the drive prevents an edge dissipator from propagating into the bulk, Eq. (50) would still hold even for a dissipator acting on the entire chain. Would property (ii) still be true in that case or would the phase diagram change?

e) Would $\gamma_{\alpha\beta}(\epsilon_m-\epsilon_n-k\omega)$ in Eq. (82) have a simple representation in terms of Feynmann diagrams of the dissipative process?

Overall the work is good and the novelty and usefulness of the method presented are sufficiently demonstrated to warrant publication in SciPost Physics Core. Thus I recommend publication of the manuscript once the issues raised in this report are properly addressed.

Requested changes

1 - Add a (brief) discussion of what would change in the results of Sec. 5.2 if the dissipation did not occur on the edges (remark d).

2 - Add an explanation of why exactly Eq. (50) implies property (ii) (remark c).

3 - Add a better connection between Appendix C and Eq. (49) (remark a).

4 - Fix the panel labels in Figure 10, 11 and 12.

5 - Improve the readability of the plot labels in Figure 3, 4 and 6.

6 - Eq. (124) should read $[[H_{m_{M+1}},H_{m_M,...m_2,m_1}],\rho]$.

7 - After Eq. (50) it should be specified that $\mathcal H_m\propto\mathcal S_z$ for $m\neq0$.

8 - (Not required) it would be nice (for my personal curiosity) if the authors could comment on remark b) and e).

---

## Round 2 · Referee Report · Anonymous (Referee 2) · 2021-10-8

Strengths

1) By giving a detailed description of how to apply van Vleck high-frequency expansions to the problem of time-periodic Lindblad equations, the authors provide a useful guide for future theoretical work exploring open Floquet systems.

2) The analysis of the conditions allowing for the approximate preparation of Gibbs-like states of Floquet systems is relevant for the idea of using thermal baths for counteracting unwanted driving-induced heating.

3) The derivation of the main results is very clear.

Weaknesses

1) After computing the effective Liouvillian and the micromotion operator using the high-frequency expansion, the non-equilibrium steady state still needs to be computed numerically by finding the eigenstate of the effective Liouvillian with eigenvalue zero.

2) The paper is very long. The sheer number of examples and the extensive review of known results on Floquet-Born-Markov theory, might distract the reader a bit from the main results.

Report

In their manuscript “Non-equilibrium steady states in the Floquet-Lindblad systems: van Vleck’s high-frequency expansion approach”, Ikeda et al. investigate steady states of open quantum systems that are described by a time-periodic Lindblad equation by employing a high-frequency expansion. For that purpose, they use the fact that the time evolution super operator can be expressed in terms of two superoperators, a time-periodic micromotion operator as well as an effective time-independent Liouvillian. They point out that the system approaches a non-equilibrium time-periodic quasi steady sate that is given by the steady state of the effective generator (which is assumed to be unique) evolved with the micromotion operator. Finally, they derive the leading terms of a high-frequency expansion of the both micromotion operator and the effective Liouvillian, starting from a general expression that was earlier derived for isolated systems using van Vleck degenerate state perturbation theory in Floquet space. The so-obtained operators, which depend on the Fourier components of the Hamiltonian and the jump-operators that define the time-periodic Lindbladian of the master equation, are then employed to numerically compute the non-equilibrium quasi steady state of the open Floquet system. To illustrate their approach, the authors apply it to a number of different concrete problems, given by a driven XY-type spin-1/2 chain with edge dissipation, a three-level model for NV centers, as well as to demonstrate the inverse Faraday effect in an open driven Heisenberg chain. In these contexts, also a modified formalism is presented, where the high-frequency approximation of the system’s Floquet states is used to microsopically derive the Floquet Lindblad equation. Finally, a detailed and discussion of the conditions for approximately finding a density matrix resembling a Gibbs states is provided.

The subject of the paper, controlling quantum systems by a combination of driving and dissipation, is timely, the proposed approach sound and interesting, and the presented analyses explained mostly in a very clear fashion. However, before I can recommend publication, I would like the authors to address the following points:

(1) I think in the abstract and the introduction it does not become sufficiently clear that the proposed high-frequency (HF) expansion does actually not directly provide the non-equilibrium steady state (NESS) of the system, but that it is rather employed for approximating the micromotion operator and the effective Liouvillian, from which the NESS still has to be computed numerically.

(2) In the paragraph following the one containing equation (25), it is stated that in case the one-cycle evolution operator has a negative real eigenvalue, both the Floquet Liouvillian L_F as well as the effective Liouvillian L_eff are not of Lindblad form. However, one should be aware that there are also situations, where L_F is not of Lindblad form, despite the fact that the one-cycle evolution operator has no negative real eigenvalue (see Refs. 64 and 65). In this case it can happen that L_eff is of Lindblad form, while L_F is not. An example for such a situation is described in a recent preprint [arXiv:2107.1005] that appeared parallel with the present manuscript and that equally uses van Vleck high frequency expansions.

(3) The proof of Lemma 1 is rather brief. I think it would be worth of giving a more detailed explanation either in the main text or an appendix.

(4) Regarding the second example of a three level system:

I am not sure whether the bath model used is meaningful. Namely, the dissipator is derived microscopically using the energy eigenstates of the undriven system. However, at the same time the Hamiltonian of the system has also a time-periodic term that should be taken into account when deriving the dissipator (as it is done in a later section).

The authors provide the high-frequency expansion of both the Hamiltonian and the micromotion operator. I think it would be interesting to see also the high-frequency corrections to the dissipator.

Minor point: One might also want to write the driving term as a numerated equation like all the other terms appearing in the Liouvillian.

(5) At the end of section 5 it is stated that the approach does not require numerical time integration, but uses only linear algebra. This is true. But one at the same time one still needs to compute an eigenstate of the effective Liouvillian, which is not necessarily a simple task. I think it would be good to include a discussion, in how far computing the steady state from L_eff can be easier than computing the steady state from time evolution. A place for such a discussion might also be the introduction.

(6) I have a few comments regarding section (6), where the authors discuss the Floquet-Lindblad equation, as it is obtained for a driven system coupled to a thermal bath by employing the Floquet-Born-Markov approximation in combination with a rotating-wave approximation.

The section is titled “time-dependent dissipator for weak thermal contact”. However, as the authors write, this problem can always be mapped to a time-independent problem in the interaction picture. And actually, the authors always treat this time-independent problem. Thus, emphasizing the time-dependence of the dissipation in the section title might be slightly misleading.

The approach pursued in this section is slightly different from the one described in the previous sections. Namely, the authors employ the high-frequency expansion to the system-Hamiltonian without dissipation, in order to compute the Floquet states, which are then used in a second step to derive the Floquet-Lindblad equation, which is then mapped to a time-independent problem. This is an interesting and valid approach, however, as far as I see, it is different from what has been described in chapter 3. I think the authors should either point out that now a different approach is derived or they should explain in more detail how this approach is related to the previous one.

Appendix D and section 6.1.1 almost entirely correspond to the results presented already in Ref. 41 and other papers. While this is pointed out at the beginning, during section 6.1.1. the authors always refer to appendix D, when presenting results, rather than referring to the original papers.

Since the manuscript is already very long, the authors might consider to shift the long section 6.1.1.. into appendix D, since it mainly reviews known results.

(7) The authors might consider removing section 6.3, containing the example of the inverse Faraday effect in an open driven Heisenberg chain, from the present manuscript and transforming it into an independent second publication. Namely, while most of the paper is focusing on a method, the high-frequency approximation for open quantum Floquet systems, this section puts a strong focuses on the physics of this particular effect. This is interesting, but after examples 1, 2 and 3 not necessarily required to illustrate the method. Moreover, in this way the paper becomes extremely long. However, this point (7) is only a suggestion.

Requested changes

See numbers (1)-(7) in the report.

---

## Editorial Decision

resubmitted